# Development of Functional Msalais Wines Rich in Amadori Compounds by Yeast Fermentation

**DOI:** 10.3390/foods14203471

**Published:** 2025-10-11

**Authors:** Jiachuan Yang, Liling Wang, Yuelin Li, Qiuye Xie, Bin Wang, Xuewei Shi, Yi He, Huilin Tan

**Affiliations:** 1Construction Group Key Laboratory of Special Agricultural Products Further Processing in Southern Xinjiang, College of Food Science and Engineering/Production, Tarim University, Alar 843300, China; 18782638980@163.com (J.Y.); 15886922981@163.com (Y.L.); m17877425543@163.com (Q.X.); 2College of Food Science, Shihezi University, Shihezi 832003, China; binwang0228@shzu.edu.cn (B.W.); shixuewei@shzu.edu.cn (X.S.); 3National R&D Center for Se-Rich Agricultural Products Processing, Hubei Engineering Research Center for Deep Processing of Green Se-Rich Agricultural Products, School of Modern Industry for Selenium Science and Engineering, Wuhan Polytechnic University, Wuhan 430023, China; yi.he@whpu.edu.cn; 4Aksu Regional Inspection and Testing Center, Aksu 843000, China; 19183467827@163.com

**Keywords:** Amadori compounds, high-performance liquid chromatography, response surface optimization, antioxidant

## Abstract

Msalais is a type of wine made by a series of processes such as boiling and fermentation from Hotan red grape juice. The Maillard reaction occurs during the boiling of the grape juice. The Amadori compound is a product of the early stage of the Maillard reaction, which has physiological activities such as antioxidation, anti-hypertension, and anti-hyperglycemia. The purpose of this study was to develop Msalais rich in Amadori compounds by utilizing the fermentative capabilities of different yeasts. The optimal fermentation process was obtained by response surface optimization, with the key parameters as follows: *Saccharomyces cerevisiae* Y4 and *Wickerhamomyces anomalus* Y2 (as the fermenting yeasts), fermentation temperature of 28 °C, fermentation time of 14 days, yeast inoculation amount of 2% (*V*/*V*), and ratio of *Saccharomyces cerevisiae* to non-*Saccharomyces cerevisiae* of 2:1. At the same time, HPLC-ELSD was used to detect Amadori compounds in the product of this optimal fermentation process. The contents of Fru-Pro and Fru-Asp in the optimal fermentation process were 0.2867 ± 0.0115 g/L and 0.0203 ± 0.0014 g/L, respectively, which were 0.0702 g/L and 0.026 g/L higher than those of commercially available commercial Msalais (0.2165 ± 0.0022 g/L and 0.0177 ± 0.0008 g/L, respectively). With the increase in the content of Amadori compounds, the antioxidant activity was significantly improved. The DPPH free radical scavenging ability was 116.37 ± 1.79 μmol Trolox/sample, which was 53.01 μmol Trolox/L sample higher than that of commercial Msalais. The ABTS free radical scavenging ability was 142.51 ± 1.98 μmol Trolox/L sample, which was 68.23 μmol Trolox/L sample higher than that of commercial Msalais. The total oxygen free radical absorption capacity was 132.74 ± 6.36 μmol Trolox/L sample, which was 60.12 μmol Trolox/L higher than that of the commercial Msalais. Compared with traditional Msalais produced by natural fermentation, the quality of Msalais fermented by specific yeasts has been significantly improved. These results provide a reliable basis for the fermentation of Msalais by specific yeasts and its quality optimization.

## 1. Introduction

Msalais is a specialty of Xinjiang. It is a delicious and mellow wine drink produced through boiling and natural fermentation. It has high medicinal value and is rich in amino acids, vitamins, glucose, iron, and other nutrients and trace elements needed by the human body. In recent years, as people’s attention to healthy eating continues to increase, Msalais has been favored for its unique taste and potential health benefits. Its unique flavor and color are derived from complex biochemical conversion processes [1]. In the traditional process, grape juice is boiled and fermented for a long time to form a dark brown and rich aroma. This process involves the deep participation of the Maillard reaction. As an early stable intermediate product of the Maillard reaction, the Amadori compound is not only the precursor of subsequent flavor substances but also has physiological activities [2] such as antioxidant activity, which may have a key impact on the storage stability and health attributes of Msalais. The Amadori compound is a carbonyl amine condensation reaction between reducing sugar substances such as glucose and fructose in fruit and vegetable raw materials as carbonyl donors and free amino groups contained in amino acids, peptides, and proteins. After Amadori rearrangement, the formation of 1-amino-1-deoxy-2-keto sugar, that is, the Amadori compound. The specific process is shown in Figure 1. Amadori compounds are generally solid, yellow, or white, easily soluble in water, methanol, and ethanol, and have no odor but are important non-volatile aroma precursors [3]. At present, there are studies on the end-stage products of the Maillard reaction in Msalais, such as acrylamide and 5-hydroxymethylfurfural [4], but the study of Amadori compounds in Msalais has not been reported. In Hotan red grape juice, before boiling, the content of proline was 284.68 mg/mL, and the content of aspartic acid was 8.86 mg/mL [5], which could provide sufficient amino acid substrates for the formation of Amadori compounds, and the two Amadori compounds were proved to have more functions, such as antibacterial, anticancer, antioxidant, etc. [6,7,8]. In order to improve the function of Msalais, Fru-Pro and Fru-Asp were selected as two Amadori compounds for subsequent research.

Amadori compounds, whose substrates, reducing sugars and amino acids, are the basic nutrients of food, have high reactivity. Therefore, Amadori compounds are widely found in tomato powder, chili powder, black garlic, and dried fruits and vegetables [9]. Amadori compounds have long been considered to have a negative impact on food quality and nutrition and human health [10]. For example, the loss of active ingredients such as amino acids and sugars in foods can lead to lower bioavailability of Amadori compounds. After being partially absorbed, they are excreted out of the body and cannot be used by organisms, which reduces the nutritional value of food [11]. The safety issues associated with Amadori compounds cannot be ignored. Overheating of Amadori compounds can produce advanced glycation end products and acrylamide [12], which may pose a health risk to diabetic patients. Therefore, the heating time needs to be strictly controlled during the enrichment of Amadori compounds. However, with the deepening of research, some Amadori compounds have been proven to have beneficial physiological effects on the human body. Yu [13] showed that Amadori compounds can inhibit cardiovascular and cerebrovascular diseases because they can effectively inhibit the activity of angiotensin-converting enzyme; Ha et al. [14] found that Amadori compounds with arginine residues can inhibit the activity of pancreatic amylase and glucosidase, thereby reducing the digestion and absorption of carbohydrates in the gastrointestinal tract, reducing the increase in postprandial blood glucose, thus playing a role in lowering blood glucose. Mossine [15] found that Fru-His can synergistically inhibit the proliferation of prostate cancer cells in vitro and in vivo, and an experimental diet supplemented with tomato paste and Fru-His can reduce the carcinogenic effect of carcinogenic rat prostate by 6 times. Therefore, more and more studies have focused on how to improve Amadori compounds by optimizing food processing technology.

At present, there have been reports on various detection methods of Amadori compounds. A classic amino acid analysis method, the post-column ninhydrin derivatization assay, has been used for the analysis of Amadori compounds [16]. However, this method has many shortcomings, such as insufficient separation, poor sensitivity, and a time-consuming process. The method of using trimethylsilane to partially derivatize sugars in the gas phase [17] has also been used for the analysis of Amadori compounds, but this method requires a long time of derivatization, and the separation process is extremely complicated due to the formation of tautomers. Yu et al. [18] used ligand exchange and scanning capillary electrophoresis to directly detect Amadori compounds by ultraviolet detection (UV). At present, the most commonly used methods for the analysis and determination of Amadori compounds are high-performance liquid chromatography, ion chromatography, etc. High-performance liquid chromatography mainly uses different types of chromatographic columns to separate Amadori compounds from other substances and then detect them with an ultraviolet detector, fluorescence detector, or evaporative light scattering detector. Most Amadori compounds have low ultraviolet absorption or no fluorescence characteristics, so it is usually necessary to introduce derivatization steps to improve the detection sensitivity [18], while evaporative light detectors respond to all substances and can detect Amadori compounds without complex pre-treatment. Li [19] synthesized and purified four Amadori compounds under aqueous conditions and used high-performance liquid chromatography–evaporative light detector to detect their purity up to 98%. High performance anion chromatography tandem pulsed amperometric detector [20] (HPAEC-PAD) and high performance anion chromatography tandem mass spectrometry (HPAEC-MS) have also been widely used in the qualitative analysis of Amadori compounds. In addition, high-performance liquid chromatography–tandem mass spectrometry has also been widely used.

Based on the above background, this study aims to develop a Msalais rich in Amadori compounds by using the fermentation performance of different yeasts. The Amadori compounds in Msalais were detected by high-performance liquid chromatography–evaporative light detector, and different yeasts were screened, and their fermentation performance was evaluated. Through a single-factor experiment and response surface optimization, the strain and optimum fermentation conditions of mixed fermentation were determined. The antioxidant capacity of the developed Msalais was studied, and the effect of the increase in Amadori compound content on the antioxidant capacity of Msalais was evaluated. These results may provide a new way for studying microorganisms in the Maillard reaction and open up the possibility for the wide application of Amadori compounds in the food industry and other industries.

## 2. Materials and Methods

### 2.1. Chemicals

Hetian were red grapes sourced from Awati County (Aawti, China); Fru-Asp (1-deoxy-1-L-aspartate-D-fructose) and Fru-Pro (1-deoxy-1-L-proline-D-fructose) at a purity > 95% was obtained from TRC Company (Toronto, ON, Canada); Chromatographic pure acetonitrile was purchased from Beijing Zeping Technology Co., Ltd. (Beijing, China); 2,3,5-triphenyltetrazolium chloride was obtained from Sinopharm Chemical Reagent Co., Ltd. (Shanghai, China); Dowex 50WX4 Hydrogen Ion Exchange Resin (200~400 mesh) was obtained from J&K Scientific (Beijing, China); ammonium formate (chromatographically pure) was obtained from Weiqi Boxing Biotechnology Co., Ltd. (Wuhan, China); Yeast genomic DNA rapid extraction kit was purchased from Beijing Solebold Technology Co., Ltd. (Beijing, China); WL nutrient agar was obtained from CoolLebo Technology Co., Ltd. (Beijing, China); phosphate buffer solution was obtained from Kaiji Biotechnology Co., Ltd. (Nanjing, China); DPPH (1,1-diphenyl-2-picrylhydrazyl), ABTS (2,2′-azino-bis-3-ethylbenzothiazoline-6-sulfonic acid), and Trolox (6-hydroxy-2,5,7,8-tetramethylchroman-2-carboxylic acid) were obtained from Yuanye Technology Co., Ltd. (Shanghai, China); fluorescein sodium and AAPH (2,2-azobis (2-methylpropylimidazole) dihydrochloride) were obtained from Aladdin Biotechnology Co., Ltd. (Shanghai, China); Angel Aroma Active Dry Yeast Powder (AQSX) was obtained from Angel Yeast Co., Ltd. (Yichang, China); other reagents in the test were of analytical purity.

### 2.2. Instruments

High Performance Liquid Chromatograph-Evaporative Light Scattering Detector purchased from Shimadzu, Japan (Kyoto, Japan); XBridge BEH Amide (5 μm, 4.6 mm × 250 mm) column purchased from Waters (Milford, MA, USA); Open-glass sand-core chromatography column (20 mm × 60 cm) was obtained from Rhoda Henghui Company (Beijing, China); Rotary evaporator from Shanghai Haozhuang Instrument Co., Ltd. (Shanghai, China); LDZX-50L Vertical High Pressure Steam Sterilizer Purchased from Shenan Medical Device Factory (Shanghai, China); GZX-9140MBE electric blower dryer was purchased from Boxun Instrument Co., Ltd. (Shanghai, China); Biomicroscope purchased from Jiangnan Yongxin Optical Co., Ltd. (Nanjing, China); Thermostatic incubator was purchased from Yiheng Technology Co., Ltd. (Shanghai, China); Biomicroplate reader obtained from Thermo Fisher Scientific (Waltham, MA, USA).

### 2.3. Preparation of Grape Juice Rich in Amadori Compounds

Hotan red grapes with more than 80% maturity, with full particles, and with no mechanical damage were selected and washed. After removing the stem and crushing, the grape juice was boiled and concentrated to a sugar content of 21 °BX after pressing and extracting the juice. The grape skin and water were boiled and concentrated to a sugar content of 21 °BX at a ratio of 2:1 (kg:L), and the oil bath was heated using a laminated pan. The specific process is shown in Figure 2. The filtered grape skin juice and grape juice were mixed and poured into a laminated pan to be boiled; the heating concentration temperature was set at 140 °C, and the final sugar content was 27 °BX. After cooling to room temperature (20–23 °C), it was frozen at −18 °C.

### 2.4. Establishment of HPLC-ELSD Method for the Detection of Amadori Compounds

#### 2.4.1. Chromatographic Conditions

According to the method of Li [19], some modifications were made: The chromatographic column was an XBridge BEH Amide (5 μm, 4.6 mm × 250 mm). The mobile phase was acetonitrile–ammonium formate = 80:20; the flow rate was 1.0 mL/min; and the column temperature was 35 °C. The injection volume was 10 μL; the evaporative light scattering detector drift tube temperature was 40 °C, the gain was 6, nitrogen was used as a carrier gas. The mobile phase was acetonitrile (phase A) and a 10 mmol/L ammonium formate aqueous solution (phase B), respectively. The gradient elution method was used for the determination. The gradient change in mobile phase A was as follows: from 0 min to 5 min, the volume fraction of phase A changed from 80% to 70%; from 5 min to 10 min, the volume fraction of phase A changed from 70% to 65%; from 10 min to 11 min, the volume fraction of phase A changed from 65% to 80%; from 11 min to 15 min, the volume fraction of phase A changed from 65% to 80%; the total time was 20 min.

#### 2.4.2. Sample Pretreatment

The 500 mL sample solution was diluted with an equal volume of ultrapure water, added with an appropriate amount of activated carbon powder to adsorb impurities such as pigments, and this was centrifuged at 6000 rpm for 10 min. The supernatant was filtered by medium-speed qualitative filter paper and allowed to stand for later use. The pretreated Dowex 50WX4 (Thermo Fisher Scientific) hydrogen ion exchange resin (200~400 mesh) was loaded into the chromatographic column, and the excess water was discharged by settling with ultrapure water to half of the column height. A 1 L diluted sample solution was injected into the chromatographic column, and after it flowed out naturally, it was washed with a large amount of ultrapure water to remove impurities such as glucose. Elution was performed with 0.2 mol/L ammonia water, and the TTC test positive eluent (each 10 mL as a group) was collected in sections until the TTC test was negative [21]. Then, the resin was washed with ultrapure water to neutral, soaked in 0.1 mol/L dilute hydrochloric acid for 30 min, washed with ultrapure water again to remove the acid solution, and the resin was recovered for backup. The collected eluent was vacuum-distilled at 55 °C for 30 min and concentrated for analysis.

#### 2.4.3. Methodological Investigation

Standard stock solution preparation and system adaptability test

The 10 mg Fru-Asp and Fru-Pro standards were accurately weighed, dissolved in ultrapure water, and diluted to 10 mL in volumetric flasks to prepare a 1000 μg/mL standard stock solution, which was stored in the dark at 4 °C. In the actual detection, the Fru-Pro standard was dissolved in ultrapure water and diluted to 500, 400, 300, 200, and 100 μg/mL, and the Fru-Asp was dissolved in ultrapure water and diluted to 5, 10, 100, 200, and 300 μg/mL. The HPLC-ELSD detection was performed according to the chromatographic conditions of Section 2.4.1, and the chromatogram was recorded.

2.Linear relationship investigation

The Fru-Pro standard was dissolved in ultrapure water and diluted to 500, 400, 300, 200, and 100 μg/mL, and the Fru-Asp was dissolved in ultrapure water and diluted to 5, 10, 100, 200, and 300 μg/mL. HPLC-ELSD detection was performed according to the chromatographic conditions of Section 2.4.1. The standard curve was drawn with the peak area as the *y*-axis and the standard concentration of Fru-Pro and Fru-Asp as the *x*-axis, and the correlation coefficient R^2^ was calculated.

3.Accuracy test

The Fru-Pro standard was dissolved in ultrapure water and diluted to 250 μg/mL, and a total of 3 Fru-Pro aqueous solutions were prepared. The Fru-Asp was dissolved in ultrapure water and diluted to 10 μg/mL. A total of 3 parts of Fru-Asp aqueous solution were prepared. Six standard solutions were detected by HPLC-ELSD according to the chromatographic conditions of Section 2.4.1. The peak area was recorded, and the relative standard deviation RSD was calculated.

4.Precision test

The Fru-Pro standard was dissolved in ultrapure water and diluted to 250 μg/mL. Fru-Asp was dissolved in ultrapure water and diluted to 10 μg/mL. HPLC-ELSD detection was performed according to the chromatographic conditions of Section 2.4.1. Each standard solution was injected repeatedly 6 times, the peak area was recorded, and the relative standard deviation RSD was calculated.

5.Repeatability test

Msalaisi from Miandu Distillery was used as the sample solution. The sample solution was pretreated according to Section 2.4.2 to obtain the sample solution to be tested. A total of 1 mL of the sample solution to be tested was taken and detected by HPLC-ELSD according to the chromatographic conditions of Section 2.4.1. The sample was injected repeatedly 6 times, the peak area was recorded, and the content of Amdori compounds and the relative standard deviation RSD were calculated according to the standard curve.

6.Stability test

The sample solution was taken and pretreated according to the method of Section 2.4.2 to obtain the sample solution to be tested. The sample solution to be tested was held at 20 °C for 0, 6, 12, 18, 24, 30, and 36 h, respectively. HPLC-ELSD detection was performed according to the chromatographic conditions of Section 2.4.1. The peak area at 0 h was used as a comparison to calculate the content changes and RSD values of Amadori compounds at different time points.

7.Determination of sample content

The sample solution was taken, and the sample solution was pretreated according to the method of Section 2.4.2 to obtain the sample solution to be tested. The 1 mL sample solution to be tested was accurately sucked and repeated three times to obtain three sample solutions to be tested. HPLC-ELSD detection was performed according to the chromatographic conditions of Section 2.4.1, and the concentrations of the two Amadori compounds were calculated according to the standard curve.

8.Standard addition recovery test

The sample solution was pretreated according to the method of Section 2.4.2 to obtain the sample solution to be tested. The sample solution to be tested was divided into two groups, one group was added with 0.5 mL 250 μg/mL Fru-Pro standard solution, and the other group was added with 0.5 mL 10 μg/mL Fru-Asp standard solution, with 6 parts in each group. HPLC-ELSD detection was carried out according to the chromatographic conditions of Section 2.4.1, and the average recovery rate and RSD value were calculated.

### 2.5. Screening of Yeast

#### 2.5.1. Isolation and Purification of Yeast

Preparation of YPD medium was as follows: glucose 2%, peptone 2%, yeast dip 1%, chloramphenicol 0.001%, and solid medium with 2% agar powder; the above material ratios are volume ratios [22].

Accurate weighing of soil and grapevine samples was conducted, followed by their placement into conical flasks containing 100 milliliters of sterile water. The samples were then subjected to rigorous shaking to ensure homogeneity. The 10^−1^ to 10^−7^ gradient dilutions were prepared sequentially, and 100 microliters of each sample was inoculated into WL medium by the dilution spread plate method. An inverted incubation at 28 °C was performed for a period of 48 h. Following this incubation, characteristic single colonies were selected and inoculated into fresh YPD medium using the streak plate method. This process was repeated until pure culture strains exhibiting consistent morphology were obtained. The appropriate amount of 30% glycerol aqueous solution was prepared and placed in a high-pressure steam sterilization pot for sterilization. After sterilization, it was taken out and cooled to room temperature (20–23 °C) and transferred to a 2 mL cryopreservation tube. The isolated strain was placed in a cryopreservation tube, shaken well, and the cryopreservation tube was stored in a refrigerator at −80 °C. The morphology of the colonies (e.g., colony size, color, surface features, etc.) was observed by a light microscope, and their characteristics were recorded [22].

#### 2.5.2. Molecular Biological Identification of Yeast

The isolated and purified suspected yeast strains were inoculated into YPD liquid medium and activated at 28 °C for 48 h. The bacteria were collected by centrifugation, and the genomic DNA was extracted by a yeast genomic DNA rapid extraction kit. PCR amplification was performed using ITS1 (5′-TCCGTAGGTGAACCTGCGG-3′ NR_171887.1) and ITS4 (5′-TCCTCCGCTTATTGATATATGC-3′ NR_168827.1) as primers. The reaction system (25 μL) was as follows: ddH_2_O 10.5 μL, Taq mix 12.5 μL, forward/reverse primers 0.5 μL, and template DNA 1 μL. PCR conditions were as follows: 94 °C for 5 min; 94 °C denaturation 30 s, 58 °C annealing 30 s, 72 °C extension 35 s, 30 cycles; and a final extension at 72 °C for 10 min [23]. The amplified products were detected by gel electrophoresis and sent to Xi’an Qingke Biotechnology Co., Ltd., (Xi’an, China) for sequencing. The sequencing results were compared with NCBI database, and the phylogenetic tree was constructed by the MEGA11 neighbor-joining method.

#### 2.5.3. Rescreening of Yeast

Preparation of yeast seed solution

The preserved yeast strains were inoculated in YPD liquid medium and cultured at 28 °C and 150 rpm for 48 h. The cultured yeast was transferred to a sterile centrifuge tube and centrifuged at 8000 rpm for 5 min. The precipitate was taken and dissolved in sterile water. The OD_600_ was measured by a microplate reader until the OD_600_ was 0.6–0.8. At this time, the yeast concentration was about 6–8 × 10^7^ cells/mL. The prepared yeast seed solution was stored at 4 °C for later use.

2.Determination of gas production performance

The yeast seed solution was inoculated into a Duchenne fermentation tube (containing YPD liquid medium) at 2% (*V*/*V*) and cultured at 28 °C for 48 h. The gas production in the fermentation tube (such as the number of bubbles, volume change) and the odor characteristics of the fermentation broth were observed and recorded regularly to evaluate the gas production capacity of the strain [22].

3.Determination of alcohol tolerance

The yeast seed solution was inoculated into YPD medium containing different concentrations of anhydrous ethanol (final concentrations were 6%, 9%, 12%, 15%, and 18% *V*/*V*) at a 2% (*V*/*V*) inoculation amount, and YPD medium without ethanol was used as a blank control. After 48 h of culture at 28 °C, the OD value at a 600 nm wavelength was measured by microplate reader to evaluate the growth of the strain at different alcohol concentrations [22].

4.Determination of acid resistance

The yeast seed solution was inoculated into YPD medium with a pH of 2.0, 2.5, 3.0, 3.5, and 4.0 (adjusted with citric acid) at a 2% (*V*/*V*) inoculation amount, and YPD medium without pH adjustment was used as a blank control. After incubation at 28 °C for 48 h, the OD value at a 600 nm wavelength was measured to analyze the growth adaptability of the strain under different acidic conditions [22].

5.Determination of sugar resistance

The yeast seed solution was inoculated into YPD medium with glucose concentrations of 300, 350, 400, 450, and 500 g/L at a 2% (*V*/*V*) inoculation amount, and YPD medium without additional glucose was used as a blank control. After incubation at 28 °C for 48 h, the OD value at a 600 nm wavelength was measured to investigate the growth characteristics of the strain in a high-glucose environment [22].

### 2.6. Msalais Fermentation Single-Factor Test and Response Surface Optimization Test

#### 2.6.1. Effect of Yeast Addition on the Content of Amadori Compounds in Msalais

Five fermentation tanks were taken, and 1 L of grape juice rich in Amadori compounds (see Section 2.4 for preparation method) was taken in the fermentation tank. In each fermentation tank, 1%, 2%, 3%, 4%, 5% yeast seed liquid (*V*/*V*) was added in turn. The ratio of *Saccharomyces cerevisiae* to non-*Saccharomyces cerevisiae* was 2:1 (the total volume of *Saccharomyces cerevisiae* and non-*Saccharomyces cerevisiae* was the volume of yeast seed solution), and the fermentation was carried out at 28 °C for 14 days (336 h). The above steps were repeated three times. After the fermentation, the content of Amadori compounds in each fermentation tank was detected. Determine the optimal yeast addition amount.

#### 2.6.2. Effect of Fermentation Temperature on the Content of Amadori Compounds in Msalais

Five fermentation tanks were taken, and 1 L of grape juice rich in Amadori compounds (see Section 2.4 for preparation method) was taken into the fermentation tank, and a 2% yeast seed solution (*V*/*V*) was added to each fermentation tank. The ratio of *Saccharomyces cerevisiae* to non-*Saccharomyces cerevisiae* was 2:1 (the total volume of *Saccharomyces cerevisiae* and non-*Saccharomyces cerevisiae* was the volume of the seed liquid). The five fermentation tanks were placed in an incubator at fermentation temperatures of 22 °C, 25 °C, 28 °C, 31 °C, and 34 °C, respectively, and fermented for 14 days (336 h), repeating the above steps three times. After the fermentation, the content of Amadori compounds in the Msalais was detected in each fermentor to determine the optimal fermentation temperature [24].

#### 2.6.3. Effect of Fermentation Time on the Content of Amadori Compounds in Msalais

Five fermentation tanks were taken, and 1 L of grape juice rich in Amadori compounds (see Section 2.4 for preparation method) was taken into the fermentation tank, and a 2% yeast seed solution (*V*/*V*) was added to each fermentation tank. The ratio of *Saccharomyces cerevisiae* to non-*Saccharomyces cerevisiae* was 2:1 (the total volume of *Saccharomyces cerevisiae* and non-*Saccharomyces cerevisiae* was the volume of the seed liquid), and the five fermentation tanks were placed at a fermentation temperature of 28 °C for 7 days (168 h), 14 days (336 h), 21 days (504 h), 25 days (600 h), and 30 days (720 h) of fermentation, and the above steps were repeated three times. After the fermentation, the content of Amadori compounds in the Msalais was detected in each fermentor to determine the optimal fermentation time.

#### 2.6.4. The Effect of the Ratio of Saccharomyces Cerevisiae to Non-Saccharomyces Cerevisiae on the Content of Amadori Compounds in Msalais

Five fermenters were selected, and 1 L of grape juice rich in Amadori compounds (preparation method is shown in Section 2.4) was taken into the fermenter. In each fermenter, a 2% yeast seed solution (*V*/*V*) was added according to the following ratios of *Saccharomyces cerevisiae* to non-*Saccharomyces cerevisiae*: 1:1, 1:2, 2:1, 1:3, and 3:1 (the total volume of *Saccharomyces cerevisiae* and non-*Saccharomyces cerevisiae* was the volume of the yeast seed solution). The fermentation temperature was 28 °C, and the fermentation was carried out for 14 days (336 h). The above steps were repeated three times. After the fermentation, the content of Amadori compounds in the Msalais was detected in each fermenter. The optimal ratio of *Saccharomyces cerevisiae* to non-*Saccharomyces cerevisiae* was determined.

#### 2.6.5. Msalais Fermentation Response Surface Test

According to the optimal condition range determined by the single-factor test, the content of Fru-Pro and Fru-Asp were used as evaluation indexes. The Box–Behnken experimental design method was used to select the amount of yeast, fermentation temperature, fermentation time, and the ratio of *Saccharomyces cerevisiae* to non-*Saccharomyces cerevisiae*. The four-factor three-level response surface design is shown in Table 1, and the optimal fermentation conditions were obtained according to the response surface optimization test.

### 2.7. Antioxidant Experiment

#### 2.7.1. Determination of DPPH Free Radical Scavenging Ability

A 25 μL sample was mixed with 200 μL of freshly prepared 0.35 mmol/L DPPH/methanol solution and placed in the dark at room temperature (20–23 °C) for 2 h. Methanol was used as a blank to determine the absorbance at 517 nm. The absorbance of the sample solution was recorded as Aa, and the absorbance of the blank was recorded as Ab. Trolox was dissolved in methanol with a concentration gradient of 0, 10, 20, 30, and 40 μg/mL [25]. The antioxidant activity of the Msalais was expressed as μmol Trolox/L sample (Trolox equivalent).(1)$$\mathrm{Scavenging}\;\mathrm{rate}\;\mathrm{of}\;\mathrm{DPPH}\;\mathrm{radical}\;(\%)=\frac{Aa-Ab}{\mathrm{Aa}}\times 100\%$$

Note: “Aa” is the absorbance of the sample solution.“Ab” is the blank absorbance of methanol.

#### 2.7.2. Determination of ABTS Free Radical Scavenging Ability

The configuration of ABTS working solution was as follows: ABTS mother liquor with a concentration of 7 mmol/L and potassium persulfate solution with a concentration of 2.45 mmol/L were mixed in an equal volume ratio and placed in the dark for 12–16 h at room temperature (20–23 °C). The configured ABTS working solution was diluted to an absorbance of 0.700 ± 0.020 for use. The 0.1 mL sample was mixed with 3.9 mL of reaction solution and placed in the dark at room temperature (20–23 °C) for 30 min, with methanol used as a blank. A total of 200 μL of the mixed solution was added to a 96-well microtiter plate, and then the absorbance was measured at a wavelength of 735 nm. The absorbance of the sample solution was recorded as Aa, and the absorbance of the blank was recorded as Ab. The concentration gradient of water-soluble vitamin E (Trolox) was 0, 10, 20, 30, and 40 μg/mL [26].(2)$$Scavenging\;rate\;of\;ABTS\;radical\;\left(\%\right)=\frac{Aa-Ab}{Aa}\times 100\%$$

Note: “Aa” is the absorbance of the sample solution.“Ab” is the blank absorbance of methanol.

#### 2.7.3. Determination of Total Oxygen Radical Absorbance Capacity (ORAC)

The sample solution, 20 μL of the Trolox standard solution, and 200 μL of fluorescein sodium solution (96 nmol/L) were added to a 96-well plate, and 20 μL phosphate buffer (100 mmol/L) was added to the blank control. After mixing, the sample was preheated at 37 °C for 20 min, and 20 μL of AAPH (153 mmol/L) solution was quickly added and mixed well. The 96-well plate was placed in a microplate reader for fluorescence intensity measurement. Determination conditions were as follows: excitation wavelength λ_excitation_ = 485 nm, absorption wavelength λ_emission_ = 535 nm, fluorescence value was measured every 2.5 min, a total of 35 cycles were used, and fluorescence intensity was recorded as f_1_, f_2_, … The f_35_ value was subjected to Trolox calibration in each test, and the ORAC value of the sample was expressed as Trolox equivalent [27].
AUC = (0.5 × f_1_/f_1_ + f_2_/f_1_ + f_3_/f_1_ + … + f_i_/f_1_ + … f_34_/f_1_ + 0.5 × f_35_/f_1_) × 2.5ΔAUC = AUC_sample_ − AUC_blank_(3)

Note: “AUC_sample_” is the calculation results of 35 fluorescence intensities of the sample solution.“AUC_blank_” is the calculation of 35 fluorescence intensities of the phosphate buffer solution.

## 3. Results and Discussion

### 3.1. Validation of HPLC-ELSD Method for the Detection of Two Amadori Compounds

#### 3.1.1. System Adaptability Results

The two Amadori compound standard solutions were injected for analysis, and high-performance liquid chromatograms of Fru-Asp and Fru-Pro standard solutions were obtained. The results are shown in Figure 3. It can be seen from Figure 3 that the peak times of Fru-Pro and Fru-Asp were 9.16 min and 11.78 min, respectively. Two Amadori compounds were detected within 20 min, and the peak shape and resolution of each target peak are good, which proves that the method can reliably detect these two Amadori compounds.

#### 3.1.2. Results of Linear Relationship Investigation

The standard curve regression equation and correlation coefficient results of the two Amadori compounds are shown in Appendix A. The standard curve regression equation of Fru-Pro was y = 1142.4x − 70,522, and the standard curve regression equation of Fru-Asp was y = 694.17x + 2227. Fru-Pro showed a good linear relationship in the mass concentration range of 100~500 μg/mL, and Fru-Asp showed a good linear relationship in the mass concentration range of 5~300 μg/mL, with a correlation coefficient of R^2^ ≥ 0.9995. This shows that Fru-Pro and Fru-Asp have a good linear relationship with the chromatographic peak area in the corresponding mass concentration range, and the detection sensitivity of this method is high.

#### 3.1.3. Accuracy Test Results

It was confirmed in Section 3.1.1 that this method can reliably detect two Amadori compounds, and ensuring the accuracy of detecting Amadori compounds at the same concentration is a key factor for subsequent research. The accuracy test results of the two Amadori compounds are shown in Appendix A. According to Appendix A, the relative standard deviations of the accuracy test results of Fru-Pro and Fru-Asp were 2.83 and 9.49%, respectively, and the relative standard deviations were all less than 10%, which proves that the accuracy of the method is good.

#### 3.1.4. Precision Test Results

In the subsequent test of the Msalais samples, the samples needed to be repeatedly measured, and the precision of the method affected the scientificity of various results. In Section 3.1.3, the accuracy of the method for the determination of different samples with the same concentration was verified. The precision test can reduce the error of the detected Amadori compound content. The precision test results of the two Amadori compounds are shown in Appendix A. It can be seen from Appendix A that the relative standard deviations of the precision test results of Fru-Pro and Fru-Asp were 4.59% and 8.72%, respectively, and the relative standard deviations were less than 10%, indicating that the instrument’s performance was relatively stable and the precision of the method was good.

#### 3.1.5. Repeatability Test Results

It was necessary to repeatedly determine the content of the Amadori compounds in the sample to reduce the error caused by the instrument and other irresistible factors. The repeatability test results of the two Amadori compounds are shown in Appendix A. It can be seen from Appendix A that the relative standard deviations of the repeatability test results of Fru-Pro and Fru-Asp were 2.67% and 8.06%, respectively, and the relative standard deviations were less than 10%, indicating that the method can accurately detect the content of Amadori compounds in the samples.

#### 3.1.6. Result of Stability Test

The stability test ensured that the sample did not change the content of the Amadori compounds due to short-term placement. The stability test results of the two Amadori compounds are shown in Appendix A. It can be seen from Appendix A that the relative standard deviations of Fru-Pro and Fru-Asp were 3.25% and 8.71%, respectively, and the relative standard deviations were less than 10%, indicating that the sample solution was stable within 36 h.

#### 3.1.7. Sample Content Determination Results

HPLC-ELSD was used to detect two Amadori compounds in Msalais-like varieties. The detection chromatograms of Fru-Pro and Fru-Asp are shown in Figure 4. The retention time of Fru-Pro was 9.16 min, and the retention time of Fru-Asp was 11.77 min, and the contents were 0.2165 ± 0.0022 g/L and 0.0185 ± 0.0008 g/L, respectively. The separation of the two substances is clear.

#### 3.1.8. Test Results of Recovery Rate

The recovery test results of the two Amadori compounds are shown in Appendix A. It can be seen from Appendix A that the average spiked recoveries of Fru-Pro and Fru-Asp were 101.21% and 100.57%, respectively. The relative standard deviations of the average recovery test results of each component were 4.97% and 5.87%, respectively. The relative standard deviations were less than 10%, indicating that the method is stable and reliable and can accurately test the Amadori compounds in the sample.

### 3.2. Preliminary Screening of Yeast

#### 3.2.1. Separation of Yeasts in Samples by Dilution Coating Plate Method

Three batches of samples were sampled in a Msalais winery. Yeasts were isolated from the mechanical surface, grape epidermis, and soil, and the dilution gradients were 10^−3^, 10^−3^, and 10^−2^, respectively. The coated plate picture is shown in Figure 5.

#### 3.2.2. Morphological Identification of Yeast

Single colonies were picked from the diluted coating plate and inoculated into the new YPD medium using the plate streaking method until a single morphological strain was purified. The colony morphology was observed under a biological microscope, and the characteristics were recorded. A total of 15 suspected yeast strains were isolated from the fruit of Hotan red grape, and these were numbered as Y1, Y2, Y4, Y12, Y16, Y17, Y18, Y25, Y29, Y41, Y61, Y67, Y72, Y91, and Y107. The colony morphology and microscopic morphology of the isolated suspected yeast strains were observed [22], as shown in Figure 6.

From Figure 6, it can be seen that the edges of the 15 colonies are neat and that the surface is mainly smooth and easy to evoke. The main morphological description is shown in Table 2. The cell morphology of strains Y12, Y16, Y18, Y25, Y67, Y91, and Y107 was spherical, Y17 and Y72 were oval, Y2, Y4, Y29, Y41, and Y61 were lemon-shaped. Among them, Y1, Y12, Y18, and Y107 were single-ended budding, and the remaining 10 strains were two-ended or multi-ended budding. Some of them formed a string of cells, which were like filaments [28]. The microscopic morphology is described in Table 3.

#### 3.2.3. Molecular Biological Identification Results of Yeast

The DNA of 15 yeast strains was extracted and amplified by PCR using primers ITS1 and ITS4. The sequences of the strains were compared by BLAST v2.16 in NCBI, and the phylogenetic tree [29] was constructed, as shown in Figure 7.

From Figure 7, it can be seen that the strains Y1, Y2, Y4, Y12, Y17, Y18, Y25, Y29, Y41, Y61, Y67, Y72, Y91, and Y107 are related to some other reference yeast strains downloaded from NCBI. Among them, strains Y1, Y4, and Y16 had high similarity with *Sacchsromyces cerevisiae*, and the similarity was 100%. Strain Y2 had high similarity with *Wickerhamomyces anomalus*, and the similarity was 100%. Strain Y12 had high similarity with *Hanseniaspora uvarum*, and the similarity was 99%. Strain Y17 had high similarity with *Pichia kudriavzevii*, and the similarity was 100%. Strain Y18 had high similarity with *Candida orthopsilosis*, and the similarity was 100%. Strain Y25 had a high similarity with *Starmerella apicola*, with a similarity of 100%. The strain Y29 had a high similarity with *Hanseniaspora guilliermondii*, and the similarity was 100%. Strain Y41 had high similarity with *Hanseniaspora vineae*, and the similarity was 100%. Strain Y61 had high similarity with *Torulaspora delbrueckii*, and the similarity was 100%. The strain Y67 had a high similarity with *Metschnikowia pulcherrima*, and the similarity was 100%. Strain Y72 had a high similarity with *Starmerella bacillaris*, with a similarity of 100%. Strain Y91 had high similarity with *Torulaspora delbrueckii*, and the similarity was 100%. Strain Y107 had a high similarity with *Lachancea thermotolerans*, with a similarity of 100%. The similarity was higher than 99%, and the 15 strains could be identified as belonging to the above strains.

### 3.3. Results of Yeast Rescreening Test

#### 3.3.1. Experimental Results of Gas Production Performance of Yeast Strains

The results of gas production performance of yeast are shown in Table 4. The largest gas production was obtained with *Saccharomyces cerevisiae* Y4, which had good precipitation and a strong wine aroma. Y1, Y16, and AQSX had more gas production and wine aroma. The better the gas production of yeast, the stronger the ability to decompose sugar to produce carbon dioxide and alcohol. The main fermentation product of *Saccharomyces cerevisiae* is alcohol, so a stronger wine aroma of its fermentation broth proves that it is more suitable for alcohol fermentation. The role of non-*Saccharomyces cerevisiae* is to produce some aroma substances such as alcohol and esters. The stronger the aroma, the more esters it produces, which can improve the flavor of Msalais. Non-*Saccharomyces cerevisiae* Y2, Y12, Y17, Y41, Y61, Y72, and Y91 had more gas production and good precipitation. The aroma of Y2 was stronger, and the fermentation broth of Y17 had no obvious aroma and less precipitation.

#### 3.3.2. Experimental Results of Alcohol Resistance of Yeast Strains

It can be seen from Table 5 that Y1, Y2, and Y4 can grow in large quantities when the alcohol volume fraction is 6% and can grow significantly when the alcohol volume fraction is 9%. Y16, Y107, and AQSX can grow in large quantities when the alcohol volume fraction is 6%. Y12, Y17, Y25, Y29, Y41 and Y72 can grow significantly when the alcohol volume fraction is 6%. Studies have shown that *Saccharomyces cerevisiae* has better alcohol resistance than non-*Saccharomyces cerevisiae* [22]. The growth and fermentation activities of most non-*Saccharomyces cerevisiae* yeasts are significantly inhibited or stopped after the increase in alcohol concentration (the biological mechanism is discussed according to the reviewer’s suggestion). Y1, Y2, Y4, Y16, Y107, AQSX, Y12, Y17, Y25, Y29, Y41, and Y72 have a certain alcohol tolerance, so they can be used for subsequent Msalais fermentation.

#### 3.3.3. Experimental Results of Acid Resistance of Yeast Strains

Yeasts Y1, Y2, Y4, Y61, and Y91 can grow in large quantities at pH 3, but Y61 and Y91 can only grow slightly at pH 2.5. Therefore, yeasts Y1, Y2, and Y4 grow best under different acidic conditions. Y12, Y16, Y17, Y18, Y25, and Y29 can grow significantly at pH 3 and pH 2.5, and their tolerance to acidic conditions is strong. Y16, Y41, Y72, Y107, and AQSX grow significantly at pH 3 and grow at pH 2.5, indicating that they have certain tolerance to acidic conditions. The pH of Msalais is usually between 3.4 and 3.8, and the main source of its acidity is tartaric acid [30]. In traditional processes, grape juice is mixed with grape skin and grape seeds before fermentation. The grape seeds release potassium ion K + and combine with tartaric acid to form potassium hydrogen tartrate precipitate, which will lead to an increase in pH value. However, Msalais is still generally acidic, so yeast with strong acid resistance is more suitable for the fermentation of Msalais

#### 3.3.4. Experimental Results of Sugar Tolerance of Yeast Strains

Yeasts Y1, Y2, Y4, Y12, Y16, Y41, Y61, Y72, Y91, Y107, and AQSX could grow on the medium with different concentrations of glucose, indicating that they had good sugar tolerance. The raw materials used in this study were boiled and concentrated grape juice, and the sugar content could reach 27–30 °BX. The higher the tolerance of yeasts to sugar, the better the growth status of yeasts in the early stage of fermentation. The selected 11 yeasts can be used for concentrated grape juice fermentation in the production of Msalais.

#### 3.3.5. Single-Yeast Fermentation Experiment

It can be seen from Figure 8 that the contents of Fru-Pro in Msalais fermented by non-*Saccharomyces cerevisiae* Y2, Y12, Y17, Y29, Y41, Y72, and Y91 were the highest. The contents of Fru-Asp in Msalais fermented by Y2, Y12, Y41, and Y72 were the highest. During the fermentation process, some non-*Saccharomyces cerevisiae* may use Amadori compounds as nutrients, resulting in a decrease in the content of Amadori compounds. Choosing non-*Saccharomyces cerevisiae* that do not use Amadori compounds as nutrients can effectively increase the content of Amadori compounds in Msalais. The contents of Fru-Pro and Fru-Asp in Msalais fermented by *Saccharomyces cerevisiae* Y4 were higher than those of Y1, Y16, and AQSX. Therefore, according to the results of the tolerance experiment and single-strain fermentation experiment, *Saccharomyces cerevisiae* Y4 and non-*Saccharomyces cerevisiae* Y2, Y12, Y41, and Y72 were selected as the original strains of mixed fermentation to ferment Msalais.

### 3.4. Single-Factor Experiment Results of Fermentation Process

#### 3.4.1. Effect of Yeast Addition on the Content of Fru-Pro and Fru-Asp

During the brewing process, the addition of yeast directly affects the fermentation kinetics and the production of metabolites. It can be seen from Figure 9a that the content of Amadori compounds reached the peak when the yeast addition amount was 2%, and then the content of Amadori compounds decreased with the increase in the yeast addition amount. The possible reason is that when the amount of yeast added was 2%, the glucose metabolism rate was balanced with the Amadori compound formation rate; when the amount of yeast was more than 2%, the content of Amadori compounds decreased due to substrate depletion, ethanol inhibition, and metabolic pathway competition [31]. This rule suggests that the amount of yeast added should be accurately controlled in the fermentation process to maximize the production of Amadori compounds and optimize the flavor of wine. The content of Amadori compounds in the mixed fermentation of Y12, Y41, Y72, and Y4 reached the peak when the amount of yeast added was 2% or 3%, but the content was lower than that of the mixed fermentation of Y2 and Y4 when the amount of yeast added was 2%. Therefore, Y2 and Y4 were selected as mixed-fermentation strains, and a 2% amount of yeast being added was the best level.

#### 3.4.2. Effect of Fermentation Temperature on the Content of Fru-Pro and Fru-Asp

It can be seen from Figure 9b that the content of Amadori compounds increased slowly when the fermentation temperature of four strains of yeast and Y4 was 22~28 °C. When the fermentation temperature was higher than 28 °C, the content decreased. The possible reason is that the increase in temperature can accelerate the metabolic rate of the yeast, causing it to produce too much alcohol and acid [32], thus inhibiting the formation of Amadori compounds by the yeast. At the same time, the yeast turns to decompose Amadori compounds after using the glucose. Comparing the content of Amadori compounds in the four yeast strains at different temperatures, the content of two Amadori compounds in Msalais fermented by Y2 and Y4 at 28 °C was the highest. Therefore, Y2 and Y4 were selected as the strains for mixed fermentation, and the fermentation temperature was 28 °C.

#### 3.4.3. Effect of Fermentation Time on the Content of Fru-Pro and Fru-Asp

It can be seen from Figure 9c that the content of Amadori compounds increased first and then decreased with the fermentation time. The reason may be that in the early stage of fermentation (0–14 days), the concentration of reducing sugars and amino acids in the the raw materials was high, which provided sufficient substrates for the formation of Amadori compounds. At this time, the saccharification and alcoholization of the yeasts had not yet been completely dominant. At this stage, the yeasts formed a large number of Amadori compounds, which accumulated Amadori compounds up to the peak. With the prolongation of the fermentation time (>14 days), reducing sugars were consumed by the yeasts, and amino acids were also metabolized by the microorganisms or involved in other reactions (such as protein decomposition), resulting in the reduction in raw materials for the synthesis of Amadori compounds [33]. At the same time, the increase in alcohol concentration in the fermentation broth inhibited the formation of Amadori compounds by the yeasts, which eventually led to a decrease in the content of Amadori compounds. Comparing the content of Amadori compounds of the four yeasts at different fermentation times, Y2 and Y4 had the highest content of two Amadori compounds in the 14d mixed fermentation of Msalais. Therefore, Y2 and Y4 were selected as the mixed-fermentation strains, and the fermentation time of 14d was the best level.

#### 3.4.4. Effect of the Ratio of Saccharomyces Cerevisiae Y4 to Non-Saccharomyces Cerevisiae Y2 on the Content of Fru-Pro and Fru-Asp

It can be seen from Figure 9d that the content of Amadori compounds was the highest when the ratio of *Saccharomyces cerevisiae* to non-*Saccharomyces cerevisiae* was 2:1, and the content was the lowest when the ratio was 1:2. The reason may be that *Saccharomyces cerevisiae* uses reducing sugars (such as glucose, fructose) as the main carbon source, has a fast metabolic rate, and can quickly generate ethanol and CO_2_. When the ratio was 2:1, S. cerevisiae dominated sugar decomposition, but the presence of non-S. cerevisiae could delay substrate depletion [34] and maintain the dynamic balance of sugars and amino acids required for the formation of Amadori compounds. If the proportion of *Saccharomyces cerevisiae* is too high (such as >2:1), excessive substrate consumption will inhibit the formation of Amadori compounds. In addition, non-*Saccharomyces cerevisiae* may decompose complex carbohydrates (such as polysaccharides) or proteins, release amino acids (such as γ-aminobutyric acid and alanine), and provide additional substrates for the Amadori reaction. If the proportion of non-*Saccharomyces cerevisiae* is too high (1:2), its metabolic activity is too strong, which may lead to premature depletion of the substrate and inhibit the formation of Amadori compounds. Comparing the content of two Amadori compounds in Msalais fermented by four yeast strains at different proportions of *Saccharomyces cerevisiae* and non-*Saccharomyces cerevisiae*, the content of two Amadori compounds in Msalais fermented by Y2 and Y4 at a ratio of 2:1 was the highest. Therefore, Y2 and Y4 were selected as strains for mixed fermentation, and the fermentation ratio of 2:1 was the best level.

### 3.5. Response Surface Optimization Experiment of Fermentation Process

#### 3.5.1. Response Surface Experimental Design and Results

According to the results of the above single-factor experiments, four fermentation process single factors (yeast addition, fermentation temperature, fermentation time, and ratio of *Saccharomyces cerevisiae* to non-*Saccharomyces cerevisiae*) affecting the content of Fru-Pro and Fru-Asp were selected for response surface design [35]. The results are shown in Table 6.

#### 3.5.2. Model Establishment and Significance Analysis

The variance analysis of the content test results of Fru-Pro and Fru-Asp was performed using the Design-Expert 13 software. The quadratic polynomial regression equation obtained by the software is Y1 (Fru-Pro) = 0.2583 − 0.0092A + 0.0083B − 0.0044C − 0.0036D − 0.0095AB + 0.0050AC + 0.0040AD + 0.0020BC − 0.0111BD + 0.0158CD − 0.0219A^2^ − 0.0393B^2^ − 0.0584C^2^ − 0.0194D^2^; Y2(Fru-Asp) = 0.0203 − 0.0013A + 0.0015B-0.0010C-0.0005D − 0.0022AB + 0.0013AC + 0.0008AD + 0.0001BC − 0.0015BD + 0.0040CD − 0.0023A^2^ − 0.0052B^2^ − 0.0089C^2^ − 0.0012D^2^. The R^2^ values of the two models were 0.9064 and 0.9650, and the R^2^_Adj_ values were 0.8128 and 0.9300, respectively. The results show that the fitting degree of the regression equation model is good, and the regression equation is representative. Therefore, the two models can better reflect the relationship between various factors and response values in the fermentation process of Msalais and predict the optimal process conditions [36]. The results of the variance analysis of the regression model are shown in Table 7 and Table 8.

From the results of Table 7 and Table 8, it can be seen that the F value of the Fru-Pro content model was 9.68, and the *p* value was less than 0.0001. The F value of the Fru-Asp content model was 27.56, and the *p* value wass also less than 0.0001. The two models were highly significant on the surface, so the validity of the model can be confirmed. At the same time, the values of the missing fit terms were 0.1384 and 1.34, respectively, and the *p* value was greater than 0.05, which was not significant, which means that the influence of unknown factors on the test was relatively small [37].

The quadratic terms A^2^, B^2^, C^2^, and D^2^ all had extremely significant effects on the content of Fru-Pro (*p* < 0.01), while the first term A and B, the interaction between AB and CD, and the quadratic terms B^2^ and C^2^ had extremely significant effects on the content of Fru-Asp, and the first term C and quadratic terms A^2^ and D^2^ showed significant effects (*p* < 0.05). Among them, the quadratic term C^2^ was the most influential factor in both models. It can be seen from the F value that the order of influence of the four factors on the Fru-Pro content was A > B > C > D, that is, the proportion of *Saccharomyces cerevisiae* to non-*Saccharomyces cerevisiae* > yeast addition > fermentation temperature > fermentation time. The order of influence on the Fru-Asp content was B > A > C > D, that is, yeast addition > *Saccharomyces cerevisiae* to non-*Saccharomyces cerevisiae* ratio > fermentation temperature > fermentation time [38].

#### 3.5.3. Response Surface Analysis

In order to explore the interaction of the four factors and the conditions for the highest content of Amadori compounds, according to the regression equation and variance analysis table, the response surface analysis diagram and contour map of each factor were drawn by the Design-Expert 13 software.

The effects of four factors on the content of Fru-ProThe response surface analysis is as follows: Figure 10(a_1_,a_2_) show that the slope of the response surface is gentle, and the contour is elliptical. When the proportion of *Saccharomyces cerevisiae* and non-*Saccharomyces cerevisiae* was low or high, the content of Fru-Pro increased first and then decreased with the increase in yeast addition, indicating that the interaction between the two was obvious. In Figure 10(b_1_,b_2_), the slope of the response surface is steep, and the contour line is oval. When the ratio of *Saccharomyces cerevisiae* to non-*Saccharomyces cerevisiae* was low or high, the content of Fru-Pro increased first and then decreased with the increase in fermentation temperature, indicating that the interaction between the two was significant. In Figure 10(c_1_,c_2_), the slope of the response surface is relatively flat, and the contour line is approximately circular. When the ratio of *Saccharomyces cerevisiae* to non-*Saccharomyces cerevisiae* was different, the content of Fru-Pro increased first and then decreased with the prolongation of fermentation time, indicating that the interaction between the two was not significant. In Figure 10(d_1_,d_2_), the slope of the response surface is steep, and the contour line is approximately circular. When the yeast addition amount was low or high, the content of Fru-Pro increased first and then decreased with the increase in fermentation temperature, indicating that the interaction between the two was not significant. In Figure 10(e_1_,e_2_), the slope of the response surface is gentle, and the contour is elliptical. When the amount of yeast added was different, the content of Fru-Pro increased first and then decreased with the increase in fermentation time, indicating that the interaction between the two was not significant. In Figure 10(f_1_,f_2_), the slope of the response surface is steep, and the contour line is oval. When the fermentation temperature was low or high, the content of Fru-Pro increased first and then decreased with the increase in fermentation time, indicating that the interaction between the two was significant.

Figure 10(g_1_,g_2_) show that the slope of the response surface is gentle, and the contour line is oval. When the ratio of *Saccharomyces cerevisiae* to non-*Saccharomyces cerevisiae* was low or high, the content of Fru-Asp increased first and then decreased with the increase in yeast addition, indicating that the interaction between the two was significant. In Figure 10(h_1_,h_2_), the slope of the response surface is steep, and the contour line is oval. When the ratio of *Saccharomyces cerevisiae* to non-*Saccharomyces cerevisiae* was low or high, the content of Fru-Asp increased first and then decreased with the increase in fermentation temperature, indicating that the interaction between the two was significant. In Figure 10(i_1_,i_2_), the slope of the response surface is relatively flat, and the contour line is approximately circular. When the ratio of *Saccharomyces cerevisiae* to non-*Saccharomyces cerevisiae* was different, the content of Fru-Asp increased first and then decreased with the prolongation of fermentation time, indicating that the interaction between the two was not significant. In Figure 10(j_1_,j_2_), the slope of the response surface is steep, and the contour line is approximately circular. When the yeast addition amount was low or high, the content of Fru-Asp increased first and then decreased with the increase in fermentation temperature, indicating that the interaction between the two was not significant. In Figure 10(k_1_,k_2_), the slope of the response surface is gentle, and the contour is elliptical. When the amount of yeast added was different, the content of Fru-Asp increased first and then decreased with the increase in fermentation time, indicating that the interaction between the two was significant. In Figure 10(l_1_,l_2_), the slope of the response surface is steep, and the contour line is oval. When the fermentation temperature was low or high, the content of Fru-Asp increased first and then decreased with the increase in fermentation time, indicating that the interaction between the two was significant.

The most significant interaction was fermentation temperature and time (Figure 10(f_1_,f_2_,l_1_,l_2_)), followed by yeast ratio and temperature (Figure 10(b_1_,b_2_,h_1_,h_2_)). When optimizing the process, the coupling conditions of temperature (28 °C) and time (14 days) should be determined first by the response surface method, and the suboptimal parameters of the ratio (2:1) and addition amount (2%) should be combined to maximize the content of Fru-Pro and Fru-Asp.

#### 3.5.4. Experimental Results of Optimal Process Conditions

The experimental data were optimized and predicted by the Design-Expert 13 software [35]. The fermentation process with the highest content of Amadori compounds was determined with the following parameters: The ratio of Saccharomyces cerevisiae to non-Saccharomyces cerevisiae is 2:1; The addition amount of yeast is 2 %; The fermentation temperature is 28 °C; The fermentation time is 14 days. Under these conditions, the content of Fru-Pro was 0.2980 g/L, and the content of Fru-Asp was 0.0196 g/L.

### 3.6. Antioxidant Activity

#### 3.6.1. DPPH Free Radical Scavenging Ability and ABTS Free Radical Scavenging Ability

A series of standard solutions with gradient changes in the concentration range of 0–40 μg/mL were prepared with Trolox as the reference standard substance, and their absorbance values were measured and linear regression equations were established [25]. The linear regression equation between DPPH free radical scavenging rate (%) and Trolox concentration (μg/mL) was y = 2.3538x − 0.4904, and the correlation coefficient was R^2^ = 0.9993. The linear regression equation between ABTS radical scavenging rate (%) and Trolox concentration (μg/mL) was y = 2.1485x − 0.382, and the correlation coefficient was R^2^ = 0.9994, The standard curve is shown in Figure 11. The DPPH and ABTS free radical scavenging abilities were expressed as Trolox equivalent [39]. The DPPH free radical scavenging rates of sample A, sample B, and sample C were 116.37 ± 1.79 μmol Trolox/sample, 63.36 ± 1.20 μmol Trolox/L sample, and 106.46 ± 3.31 μmol Trolox/L sample, respectively. Compared with sample B and sample C, the DPPH free radical scavenging ability of sample A increased by 53.01 μmol Trolox/L sample and 9.91 μmol Trolox/L sample, respectively. The ABTS free radical scavenging rates were 142.51 ± 1.98 μmol Trolox/sample, 74.28 ± 2.11 μmol Trolox/L sample, and 121.22 ± 1.93 μmol Trolox/L sample, respectively. Compared with sample B and sample C, the ABTS free radical scavenging ability of sample A increased by 68.23 μmol Trolox/L sample and 21.29 μmol Trolox/L sample, respectively. During the boiling process of grape juice, as the boiling time becomes longer, the content of vitamin C, reducing sugar, soluble protein, flavonoids, and other substances in the grape juice is significantly reduced [40]. Vitamin C and flavonoids are easily oxidized and decomposed, and the content decreases significantly during heating [41]. The content of reducing sugar increases first and then decrease. The reason may be that heat treatment leads to the precipitation of soluble sugar in the grape juice, and polysaccharides also hydrolyze into monosaccharides, thereby increasing the content of reducing sugar in the grape juice. Subsequently, reducing sugar is added to the Maillard reaction, and the reaction rate is higher than the formation rate of reducing sugar, resulting in a decrease in reducing sugar content [42]. Protein is hydrolyzed into peptide chains and free amino acids due to heat treatment, resulting in a decrease in its content [43]. At the same time, the generated free amino acids undergo a Maillard reaction with reducing sugars. Therefore, it can be inferred that the above substances are not the factors that enhance the antioxidant capacity of Msalais, and the significant increase in the content of Amadori compounds during cooking can be used as the main factor for the improvement of antioxidant capacity.

#### 3.6.2. Determination of Total Oxygen Free Radical Reduction Capacity (ORAC)

A series of standard solutions with gradient changes in the concentration range of 10–50 μg/mL were prepared with Trolox as the reference standard substance [44]. The absorbance values of 35 cycles were measured, and the ΔAUC was calculated according to the formula to establish a linear regression equation. The calculation results of AUC and ΔAUC of three different samples and standards are shown in Table 9. The linear regression equation between ΔAUC and Trolox concentration (μg/mL) was y = 0.1925x − 0.5923; R^2^ = 0.9997. The total oxygen radical reduction abilities of sample A, sample B, and sample C were 132.74 ± 6.36 μmol Trolox/L sample, 72.62 ± 8.19 μmol Trolox/L sample, and 100.29 ± 9.38 μmol Trolox/sample. Compared with sample B and sample C, the total oxygen free radical reduction ability of sample A increased by 60.12 μmol Trolox/L sample and 32.45 μmol Trolox/L sample, respectively.

## 4. Conclusions

In this study, we established an HPLC-ELSD method for the detection of Fru-Pro and Fru-Asp, screened a *Saccharomyces cerevisiae* Y4 and a non-*Saccharomyces cerevisiae* Y2, and verified the feasibility of the application of the two yeasts in the fermentation of Msalais. In order to obtain the best fermentation process of Msalais, the fermentation process optimized by the response surface method was as follows: fermentation temperature of 28 °C, fermentation time of 14 days, ratio of *Saccharomyces cerevisiae* Y4 to non-*Saccharomyces cerevisiae* Y2 of 2:1, and yeast inoculation amount of 2% (*V*/*V*). The contents of Fru-Pro and Fru-Asp were significantly increased by comparing Msalais with an unoptimized fermentation process, which was the same as the results of Li’s study [45]. By comparing the antioxidant properties of self-made Msalais and commercial Msalais, the DPPH scavenging ability, ABTS scavenging ability, and ORAC of the self-made Msalais were significantly improved. The traditional fermentation method of Msalais is natural fermentation. Natural fermentation is a complex process involving the interaction of multiple microorganisms, which has instability and potential health risks. The yeast screened in this study can not only increase the content of Amadori compounds in Msalais but also improve the stability of the fermentation process. Compared with natural fermentation, the fermentation process is more convenient to control and eliminate potential health risks. Therefore, the fermentation of Msalais by specific yeasts has great application prospects. The results of this study can also provide a theoretical basis for the development of new products.

In this study, although the Msalais process optimization system was initially constructed and the directional enrichment of the target components was achieved, there are still many directions worthy of further exploration: only two Amadori compounds, Fru-Pro and Fru-Asp, were analyzed, and other Amadori compounds and other Maillard reaction products were not analyzed. The research has limitations. In the future, non-targeted methods (such as HPLC-MS and NMR) can be used to analyze Amadori compounds more comprehensively and to explore potential volatile components and toxic by-products. The regulation mechanism of key enzymes (such as aldose reductase, glycosyltransferase, etc.) in the dynamic process of the Maillard reaction on the formation of Amadori compounds can be further analyzed by molecular biology methods. Combined with metabolomics and sensory omics technology, the correlation model between Amadori compounds and Msalais flavor quality was established to achieve accurate regulation of product quality.

## Figures and Tables

**Figure 1 foods-14-03471-f001:**
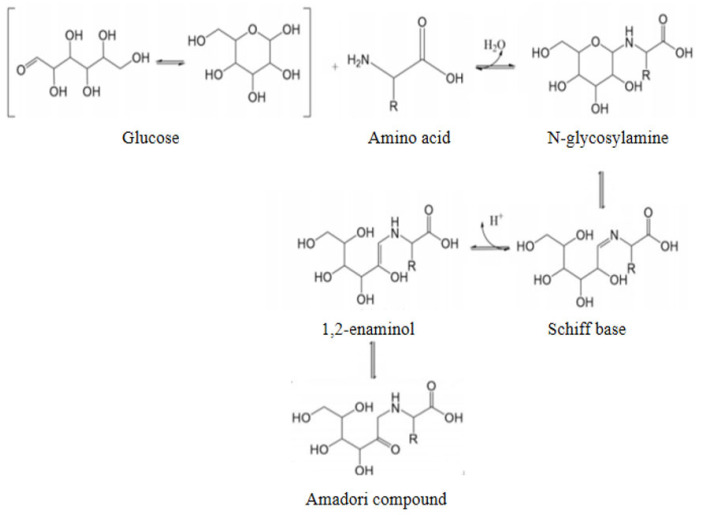
The formation of Amadori compounds.

**Figure 2 foods-14-03471-f002:**
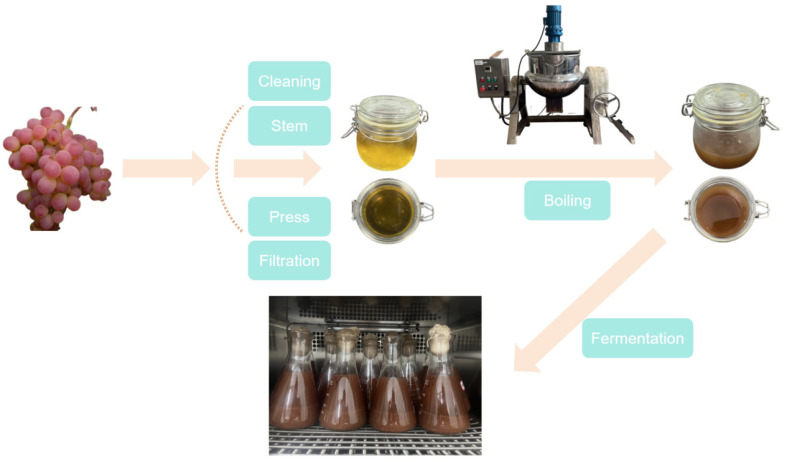
The process of making grapes into Msalais.

**Figure 3 foods-14-03471-f003:**
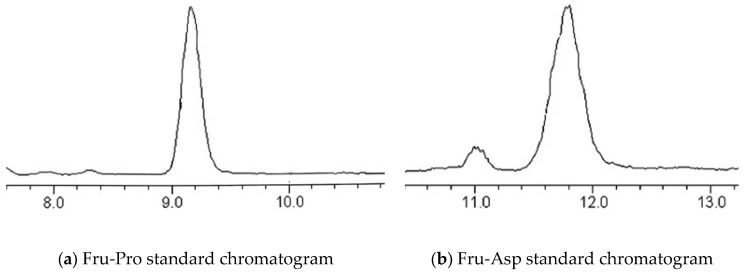
Standard chromatograms of two Amadori compounds.

**Figure 4 foods-14-03471-f004:**
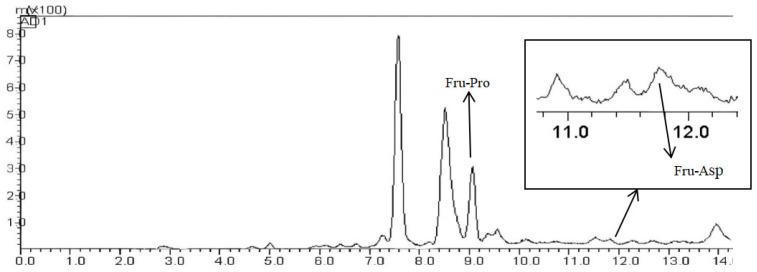
Chromatogram of two Amadori compounds.

**Figure 5 foods-14-03471-f005:**
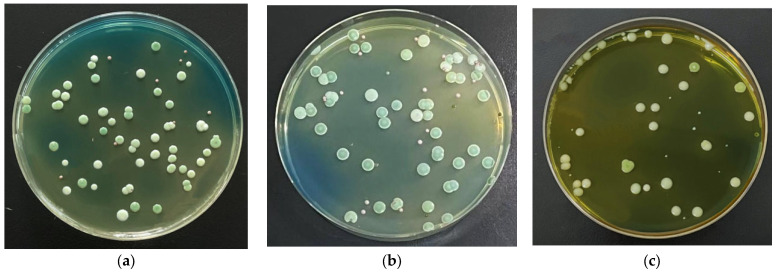
Screening of yeasts by dilution coating plate method. (**a**) Sample coating on mechanical surface, (**b**) Sample coating of grape epidermis, (**c**) Sample coating of soil.

**Figure 6 foods-14-03471-f006:**
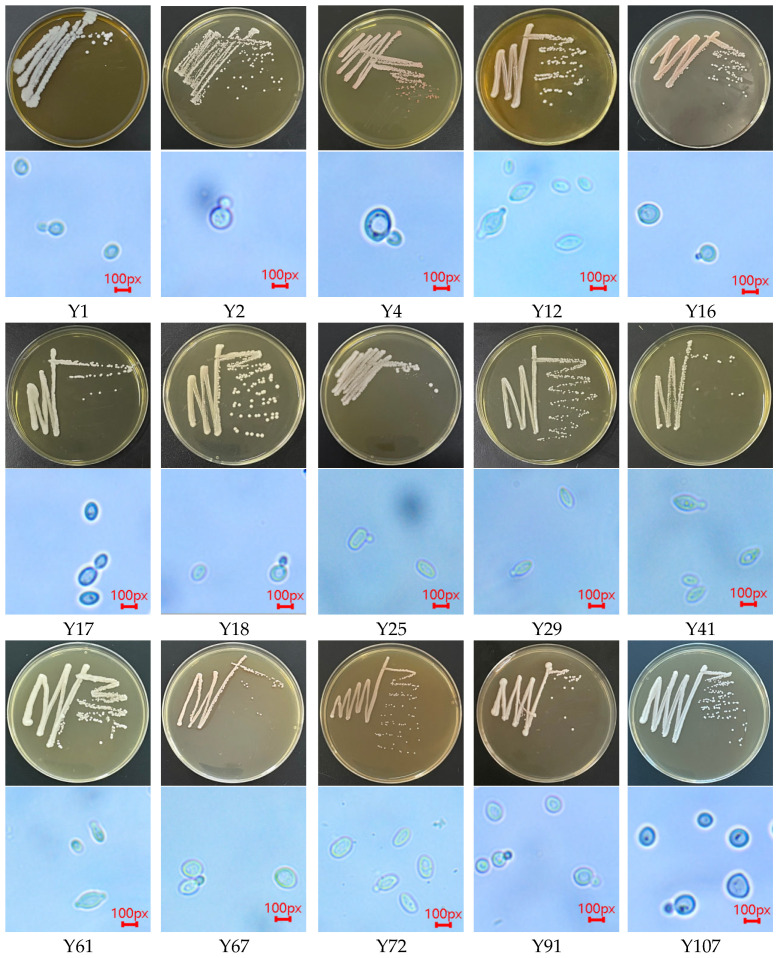
Colony morphology and microscopic morphology of 15 yeast strains.

**Figure 7 foods-14-03471-f007:**
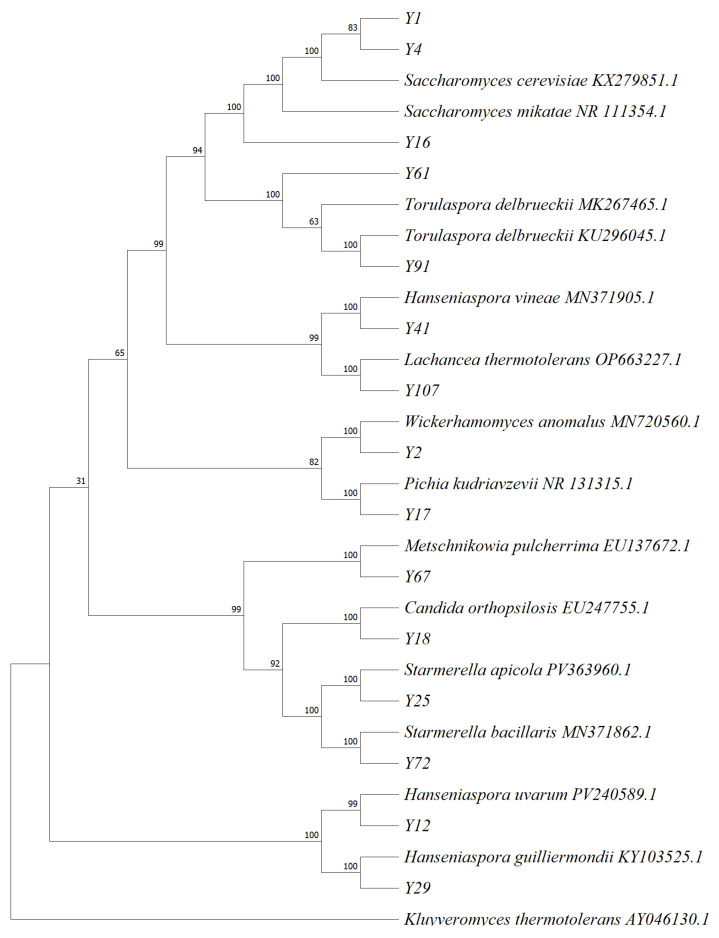
Phylogenetic tree of 15 yeast strains.

**Figure 8 foods-14-03471-f008:**
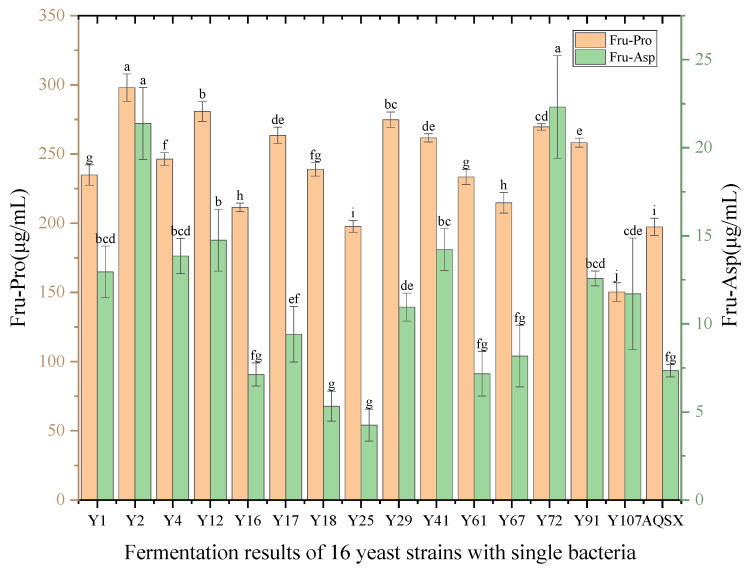
The content of Fru-Pro and Fru-Asp in Msalais fermented by 16 yeasts. Different letters indicate that the corresponding values are significantly different at different temperatures (*p* < 0.05).

**Figure 9 foods-14-03471-f009:**
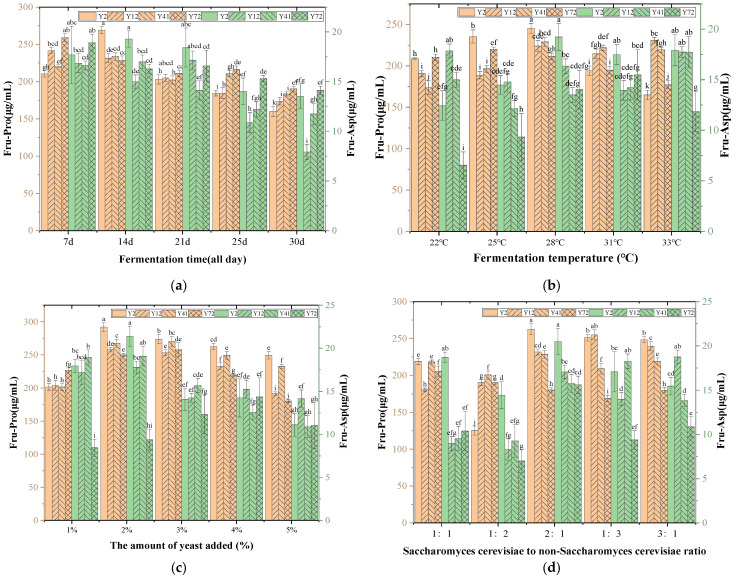
The content of Fru-Pro and Fru-Asp in Msalais fermented by 16 yeasts. (**a**) Effect of yeast addition on the content of Fru-Pro and Fru-Asp. (**b**) The content of two Amadori compounds fermented by four yeast strains at different fermentation temperatures. (**c**) The content of two Amadori compounds fermented by 4 yeast strains at different fermentation times. (**d**) The content of two Amadori compounds in Msalais fermented by four yeast strains at different ratios. Different letters indicate that the corresponding values are significantly different at different temperatures (*p* < 0.05).

**Figure 10 foods-14-03471-f010:**
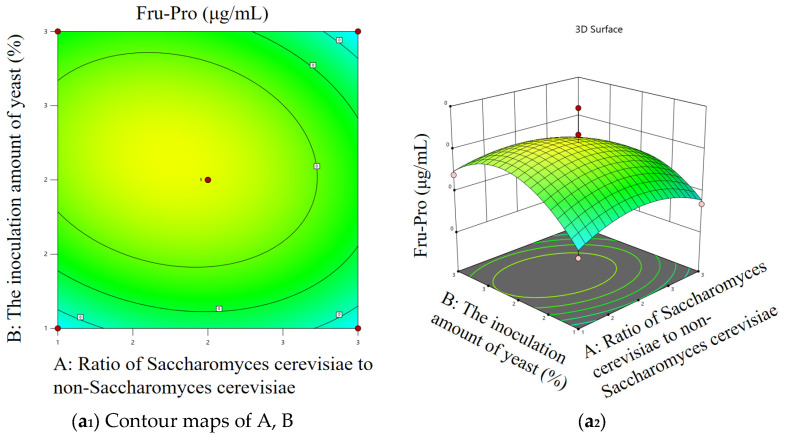
The response surface diagram and contour map of the interaction between the four factors on the content of Fru-Pro and Fru-Asp. (**a_1_**–**l_1_**) represents the contour map of response surface optimization, and (**a_2_**–**l_2_**) represents the surface map of response surface optimization.

**Figure 11 foods-14-03471-f011:**
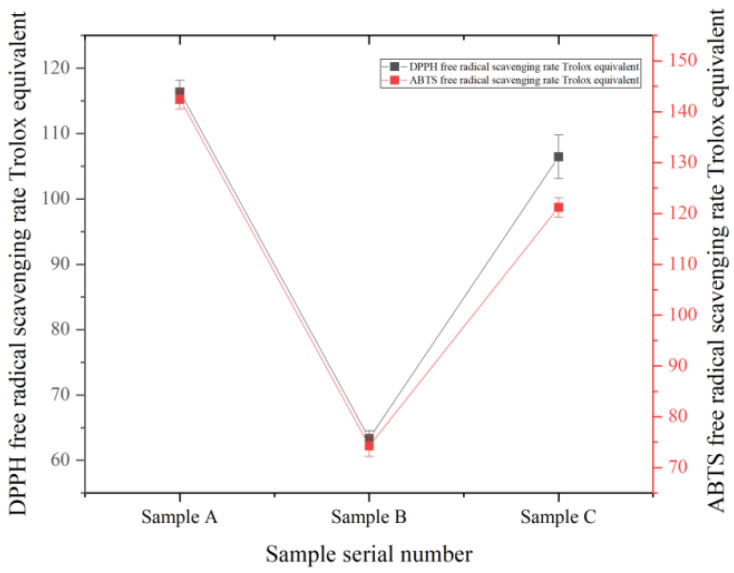
DPPH free radical scavenging ability of three different samples.

**Table 1 foods-14-03471-t001:** Response surface test factors and levels.

Level	Factor			
	A The Ratio of *Saccharomyces cerevisiae* to Non-*Saccharomyces cerevisiae*	B Yeast Addition Amount (%)	C Fermentation Temperature (°C)	D Fermentation Time (All Day)
−1	1:1	1	25	7
0	2:1	2	28	14
1	3:1	3	31	21

**Table 2 foods-14-03471-t002:** Description of colony morphology.

Number of Strains	Colony Color and Morphology
Y1, Y2	Milk white–greenish, spherical protrusions, smooth surface, opaque
Y4, Y16, Y18, Y67, Y72, Y91	Rice white–purple, spherical protrusions, rough surface, opaque
Y12, Y61	Rice white–yellowish, spherical protrusions, smooth surface, translucent
Y17	Milk white–greenish, spherical protrusions, rough surface, opaque
Y25	Milk white–greenish, flat, smooth surface, translucent
Y29	Rice white–greenish, spherical protrusions, smooth surface, translucent
Y41, Y107	Milk white, spherical protrusions, smooth surface, opaque

**Table 3 foods-14-03471-t003:** Microscopic morphology description table.

Number of Strains	Micro-Morphology
Y1	Spindle, single-ended budding reproduction
Y2, Y4, Y29, Y41, Y61	Lemon-shaped, both ends of the bud reproduction
Y12, Y16, Y18, Y25, Y67, Y91, Y107	Spherical, single-ended budding reproduction
Y17, Y72	Oval, single-ended budding reproduction

**Table 4 foods-14-03471-t004:** Experimental results of gas production performance of yeast strains.

Number of Strains	Gas Production Situation	Coagulation of Strains	Fermentation Broth Aroma
Y1	+++	The precipitation is in good condition	Wine flavor
Y2	+++	The precipitation is in good condition	Stronger aroma
Y4	++++	The precipitation is in good condition	Stronger wine aroma
Y12	+++	The precipitation is in good condition	Wine flavor
Y16	+++	The precipitation is in good condition	Wine flavor
Y17	+++	Precipitation is poor	No aroma
Y18	++	The precipitation is in good condition	No aroma
Y25	++	The precipitation is in good condition	Wine flavor
Y29	+	The precipitation is in good condition	Wine flavor
Y41	+++	The precipitation is in good condition	Lighter aroma
Y61	+++	The precipitation is in good condition	Wine flavor
Y67	++	The precipitation is in good condition	Wine flavor
Y72	+++	The precipitation is in good condition	Lighter aroma
Y91	+++	The precipitation is in good condition	Wine flavor
Y107	++	The precipitation is in good condition	No aroma
AQSX	+++	The precipitation is in good condition	Wine flavor

Note: “+” indicates gas production, and “++”, “+++” and “++++” symbols indicate greater gas production.

**Table 5 foods-14-03471-t005:** Results of alcohol, acid and sugar tolerance test of yeasts.

Various Tolerance Names		Number of Strains															
		Y1	Y2	Y4	Y12	Y16	Y17	Y18	Y25	Y29	Y41	Y61	Y67	Y72	Y91	Y107	AQSX
Medium alcohol volume fraction (%)	6	+++	+++	+++	++	+++	++	+	++	++	++	+	*	+	+	+++	+++
	9	++	++	++	*	+	+	*	+	*	*	*	*	*	*	+	+
	23	*	*	*	*	*	*	*	*	*	*	*	*	*	*	*	*
	15	*	*	*	*	*	*	*	*	*	*	*	*	*	*	*	*
	18	*	*	*	*	*	*	*	*	*	/	/	*	*	*	*	*
pH	2	*	*	*	*	*	*	*	*	*	*	*	*	*	*	*	*
	2.5	+	+	+	++	+	++	++	++	++	+	*	+	+	*	+	*
	3	+++	+++	+++	++	++	++	++	++	++	++	+++	+	++	+++	++	++
	3.5	+++	+++	+++	+++	+++	++	+++	+++	++	++	+++	++	++	+++	+++	+++
	4	+++	+++	+++	+++	+++	++	+++	+++	+++	++	+++	++	+++	+++	+++	+++
Medium sugar concentration (g/L)	300	+++	+++	+++	++	+++	+++	+++	+++	++	+++	+++	++	+++	+++	+++	+++
	350	+++	+++	+++	++	+++	+++	+++	+++	++	+++	+++	++	+++	+++	+++	+++
	400	+++	+++	+++	++	+++	++	++	++	++	++	+++	+	++	+++	+++	+++
	450	+++	+++	+++	++	+++	++	++	++	++	++	+++	+	++	+++	+++	+++
	500	+++	+++	+++	++	+++	++	++	++	++	++	+++	+	++	+++	+++	+++

Note: “/”: no growth, OD_600_ ≤ 0.1; “*”: micro-growth, 0.1 < OD_600_ < 0.3; “+”: growth, 0.3 < OD_600_ < 0.5; “++”: obvious growth, 0.5 < OD_600_ < 1.0; “+++”: a large level of growth, OD_600_ > 1.0.

**Table 6 foods-14-03471-t006:** Response surface experimental design and results.

Serial Number	A The Ratio of *Saccharomyces cerevisiae* to Non-*Saccharomyces cerevisiae*	B Yeast Inoculation Amount (%)	C Fermentation Temperature (°C)	D Fermentation Time (All Day)	Fru-Pro (g/L)	Fru-Asp (g/L)
1	−1	0	0	−1	0.2351	0.0185
2	−1	0	0	1	0.2342	0.0183
3	1	1	0	0	0.1862	0.0114
4	1	0	0	1	0.2154	0.0175
5	0	1	−1	0	0.1712	0.0078
6	−1	0	1	0	0.1760	0.0083
7	0	0	0	0	0.2389	0.0190
8	1	0	1	0	0.1723	0.0081
9	0	0	0	0	0.2435	0.0191
10	0	0	1	1	0.1818	0.0107
11	0	−1	0	1	0.1918	0.0131
12	1	0	−1	0	0.1708	0.0075
13	0	0	1	−1	0.1599	0.0052
14	0	0	−1	1	0.1603	0.0058
15	0	−1	−1	0	0.1662	0.0063
16	0	0	0	0	0.2438	0.0201
17	−1	−1	0	0	0.1799	0.0088
18	1	0	0	−1	0.2002	0.0144
19	0	1	0	1	0.1896	0.0127
20	0	1	1	0	0.1673	0.0071
21	−1	0	−1	0	0.1946	0.0131
22	−1	1	0	0	0.2198	0.0180
23	1	−1	0	0	0.1845	0.0109
24	0	0	−1	−1	0.2018	0.0163
25	0	−1	0	−1	0.1884	0.0124
26	0	0	0	0	0.2674	0.0213
27	0	−1	1	0	0.1543	0.0045
28	0	0	0	0	0.2980	0.0218
29	0	1	0	−1	0.2307	0.0180

**Table 7 foods-14-03471-t007:** Analysis of variance for Fru-Pro regression model.

Source of Variance	Quadratic Sum	Degree of Freedom	Mean Square	F Ratio	*p* Value	Significance
model	0.0321	14	0.0023	9.68	<0.0001	**
A	0.0010	1	0.0010	4.27	0.0578	
B	0.0008	1	0.0008	3.50	0.0826	
C	0.0002	1	0.0002	0.9992	0.3345	
D	0.0002	1	0.0002	0.6503	0.4335	
AB	0.0004	1	0.0004	1.54	0.2351	
AC	0.0001	1	0.0001	0.4263	0.5244	
AD	0.0001	1	0.0001	0.2735	0.6092	
BC	0.0000	1	0.0000	0.0675	0.7988	
BD	0.0005	1	0.0005	2.09	0.1703	
CD	0.0010	1	0.0010	4.24	0.0586	
A^2^	0.0031	1	0.0031	13.11	0.0028	**
B^2^	0.0100	1	0.0100	42.21	<0.0001	**
C^2^	0.0222	1	0.0222	93.51	<0.0001	**
D^2^	0.0024	1	0.0024	10.26	0.0064	**
Residual	0.0033	14	0.0002			
Lack of Fit	0.0009	10	0.0001	0.1384	0.9947	
Pure Error	0.0025	4	0.0006			
Cor Total	0.0354	28				
R^2^	0.9064					
R^2^_Adj_	0.8128					

Note: “**” indicates that the difference was extremely significant (*p* < 0.01).

**Table 8 foods-14-03471-t008:** Analysis of variance for Fru-Asp regression model.

Source of Variance	Quadratic Sum	Degree of Freedom	Mean Square	F Ratio	*p* Value	Significance
model	0.0008	14	0.0001	27.56	<0.0001	**
A	0	1	0	9.4	0.0084	**
B	0	1	0	13.83	0.0023	**
C	0	1	0	6.11	0.0268	*
D	3.52 × 10^−6^	1	3.52 × 10^−6^	1.76	0.2053	
AB	0	1	0	9.48	0.0082	**
AC	6.76 × 10^−6^	1	6.76 × 10^−6^	3.39	0.087	
AD	2.72 × 10^−6^	1	2.72 × 10^−6^	1.36	0.2623	
BC	6.25 × 10^−8^	1	6.25 × 10^−8^	0.0313	0.8621	
BD	8.41 × 10^−6^	1	8.41 × 10^−6^	4.21	0.0593	*
CD	0.0001	1	0.0001	32.07	<0.0001	**
A^2^	0	1	0	16.59	0.0011	*
B^2^	0.0002	1	0.0002	86.52	<0.0001	**
C^2^	0.0005	1	0.0005	260.18	<0.0001	**
D^2^	0	1	0	5.05	0.0412	*
Residual	0	14	2.00 × 10^−6^			
Lack of Fit	0	10	2.15 × 10^−6^	1.34	0.4169	
Pure Error	6.41 × 10^−6^	4	1.60 × 10^−6^			
Cor Total	0.0008	28				
R^2^	0.9650					
R^2^_Adj_	0.9300					

Note: “*” indicates significant difference (*p* < 0.05); “**” indicates that the difference was extremely significant (*p* < 0.01).

**Table 9 foods-14-03471-t009:** ORAC of 3 different samples.

Name	AUC	ΔAUC	Trolox Equivalent
Trolox concentration 10 μg/mL	84.83	1.29	\
Trolox concentration 20 μg/mL	86.36	3.33	\
Trolox concentration 30 μg/mL	87.74	5.13	\
Trolox concentration 40 μg/mL	89.08	7.15	\
Trolox concentration 50 μg/mL	90.74	9.01	\
Sample A	87.29 ± 0.31	5.80 ± 0.31	132.74 ± 6.36
Sample B	84.40 ± 0.39	2.91 ± 0.39	72.62 ± 8.19
Sample C	85.73 ± 0.45	4.24 ± 0.45	100.29 ± 9.38
Blank sample	81.49	\	\

## Data Availability

The original contributions presented in this study are included in the article and Appendix A. Further inquiries can be directed to the corresponding author.

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
