# Peer review of "Development of Functional Msalais Wines Rich in Amadori Compounds by Yeast Fermentation"

_foods, 2025, doi:10.3390/foods14203471_

Round 1
Reviewer 1 Report (Previous Reviewer 1)
Comments and Suggestions for Authors
The authors have revised the manuscript based on the reviewer's comments, and it now meets the standards for publication in the journal Foods. I recommend acceptance.
Author Response
Comments 1: The authors have revised the manuscript based on the reviewer's comments, and it now meets the standards for publication in the journal Foods. I recommend acceptance.
Response 1: We sincerely appreciate your positive evaluation of our manuscript and your recommendation for acceptance in the journal Foods. Your careful review and recognition of the study’s significance have been a great encouragement to us.
Reviewer 2 Report (Previous Reviewer 2)
Comments and Suggestions for Authors
I would like to thank the authors for their comprehensive and thoughtful responses to my initial comments. The revisions have significantly improved the clarity, methodological transparency, and overall scientific rigor of the manuscript. Important improvements that strengthen the study include the justification for compound selection, clarification of replicates, inclusion of statistical analyses, and comparative antioxidant data. The manuscript is now much clearer and closer to being ready for publication.
- Although many corrections have been made, the readability can be improved further by shortening some overly long sentences.
-
A few residual issues remain, for example:
-
Line 159: "( 5 μm, 4.6 mm × 250 mm )" should be corrected to "(5 μm, 4.6 mm × 250 mm)", line 187: "6000rpm for 10min" should be formatted consistently as "6000 rpm for 10 min", line 161, 196, 198, 204, 212-213, 218, 220, 240, 254-255, 267, 285, 298-300, 307-308, etc.: Please check carefully for similar minor formatting inconsistencies, particularly spacing between numbers and units.
-
Temperature formatting "20-23°C" should be written as "20–23 °C"
-
Section title: "4. conclusion" should be corrected to "4. Conclusions"
-
Figures 11 and 12: Text is too small and not easily readable; font size should be increased.
-
- The authors acknowledge the study’s limitations, which include being restricted to two Amadori compounds and excluding volatile or late-stage Maillard products. However, a brief expansion on the broader implications of Amadori compound enrichment in foods, particularly with respect to potential sensory impacts and consumer safety considerations, would benefit the discussion.
Addressing these points will improve the clarity and presentation of the manuscript further. After making these minor revisions, I believe the paper will be ready for acceptance.
Author Response
Comments 1: Although many corrections have been made, the readability can be improved further by shortening some overly long sentences.
Response 1: Thank you for pointing this out, I agree with this comment, so I have deleted some too long sentences and simplified many languages. These parts have been marked out in the article.
Comments 2: Line 159: "( 5 μm, 4.6 mm × 250 mm )" should be corrected to "(5 μm, 4.6 mm × 250 mm)", line 187: "6000rpm for 10min" should be formatted consistently as "6000 rpm for 10 min", line 161, 196, 198, 204, 212-213, 218, 220, 240, 254-255, 267, 285, 298-300, 307-308, etc.: Please check carefully for similar minor formatting inconsistencies, particularly spacing between numbers and units. Temperature formatting "20-23°C" should be written as "20–23 °C". Section title: "4. conclusion" should be corrected to "4. Conclusions". Figures 11 and 12: Text is too small and not easily readable; font size should be increased.
Response 2: Thank you for pointing this out, I agree with this comment, so I have modified the spacing of all numbers and units, modified the spelling errors, and corrected the problem that the fonts in Figure 10 and Figure 11 are too small. All the modified parts have been marked in the article.
Comments 3:The authors acknowledge the study’s limitations, which include being restricted to two Amadori compounds and excluding volatile or late-stage Maillard products. However, a brief expansion on the broader implications of Amadori compound enrichment in foods, particularly with respect to potential sensory impacts and consumer safety considerations, would benefit the discussion.
Response 3: Thank you for pointing this out, I agree with this comment, so I have extended the discussion on the safety of Amadori compounds in lines 53, 78 and 861.
Reviewer 3 Report (New Reviewer)
Comments and Suggestions for Authors
Comments:
-
The document is too extensive; it is necessary to synthesize and organize the entire article so that the reader does not lose focus on the research and to avoid confusion.
-
Review all spacing between numbers and units throughout the document.
-
The abstract does not end with a conclusion section; it is necessary to add one.
-
Add spacing between the main text and the references.
-
Review the spacing used in concentration units.
-
Review the spacing between periods in each sentence.
-
In the methodology section, it is necessary to mention the different treatments of the entire experiment.
-
Add figure captions for the equations, explaining the meaning of each variable.
-
The document contains too many figures (14), and some of them are overloaded with graphics. It is necessary to select the most relevant ones to avoid confusing the reader, or move some into supplementary material.
-
The document contains too many tables (17); it is necessary to select the most relevant ones to avoid confusing the reader, or move some into supplementary material.
-
Figure 10 is overloaded; the presentation needs to be improved.
-
The conclusion section reads more like a summary. It should highlight the main findings and close with the significance of the study.
-
The results obtained must be better discussed and interlinked across the subsections of the “Results and Discussion” section.
-
Improve the overall order and sequence of the entire document.
Author Response
Comments 1: The document is too extensive; it is necessary to synthesize and organize the entire article so that the reader does not lose focus on the research and to avoid confusion.
Response 1: Thank you for pointing this out, I agree with this comment, so I have condensed and integrated the content of the article, and the specific modifications have been marked in the article. At the same time, I have streamlined the language and deleted the redundant language that is not related to the research content.
Comments 2: Review all spacing between numbers and units throughout the document.
Response 2: Thank you for pointing this out, I agree with this comment, so I have added spacing between numbers and units, and the specific changes have been marked out in the article
Comments 3: The abstract does not end with a conclusion section; it is necessary to add one.
Response 3: Thank you for pointing this out, I agree with this comment, so I modified the end of the abstract section and added the conclusion, the specific changes have been marked out in the article
Comments 4: Add spacing between the main text and the references.
Response 4: Thank you for pointing this out, I agree with this comment, so I have added a space between the text and the references, and the specific changes have been marked out in the article
Comments 5: Review the spacing used in concentration units.
Response 5: Thank you for pointing this out, I agree with this comment, so I have modified the spacing of concentration units. The specific modification has been marked in the article.
Comments 6: Review the spacing between periods in each sentence.
Response 6: Thank you for pointing this out, I agree with this comment, so I modified the spacing of the sentence points, the specific changes have been marked out in the article
Comments 7: In the methodology section, it is necessary to mention the different treatments of the entire experiment.
Response 7: Thank you for pointing this out, I agree with this comment, in the method part, I modified and marked different processing methods.
Comments 8: Add figure captions for the equations, explaining the meaning of each variable.
Response 8: Thank you for pointing this out, I agree with this comment, so I have added comments to the equation in line 392,406,419, and the specific changes have been marked out in the article
Comments 9:The document contains too many figures (14), and some of them are overloaded with graphics. It is necessary to select the most relevant ones to avoid confusing the reader, or move some into supplementary material.
Response 9: Thank you for pointing this out, I agree with this comment, so I have put some of the figures into the supplementary material, and the remaining figures are necessary to explain the research content.
Comments 10:The document contains too many tables (17); it is necessary to select the most relevant ones to avoid confusing the reader, or move some into supplementary material.
Response 10: Thank you for pointing this out, I agree with this comment, so I have put some of the tables into the supplementary materials, and merged some of the tables together. The remaining tables are necessary to explain the research content.
Comments 11: Figure 10 is overloaded; the presentation needs to be improved.
Response 11: Thank you for pointing this out, I agree with this comment, so I have modified Figure 10 and other overloaded images and enlarged the font to make it clearer.
Comments 12: The conclusion section reads more like a summary. It should highlight the main findings and close with the significance of the study.
Response 12: Thank you for pointing this out, I agree with this comment, so I have revised the conclusion section to highlight the importance of the research content and streamlined the language.
Comments 13: The results obtained must be better discussed and interlinked across the subsections of the “Results and Discussion” section.
Response 13: Thank you for pointing this out, I agree with this comment, so I have revised the conclusion section to highlight the importance of the research content and streamlined the language.
Comments 14: The results obtained must be better discussed and interlinked across the subsections of the “Results and Discussion” section.
Response 14: Thank you for pointing this out, I agree with this comment, so I have modified the order of the method part and the result part to make the content of the article more reasonable. The modified part has been marked in the article.
This manuscript is a resubmission of an earlier submission. The following is a list of the peer review reports and author responses from that submission.
Round 1
Reviewer 1 Report
Comments and Suggestions for Authors
The manuscript addresses an interesting topic at the intersection of fermentation technology and food chemistry, focusing on the optimization of fermentation conditions for producing Msalais wine enriched in Amadori compounds. Although the subject has promising application potential, the manuscript contains several shortcomings and deficiencies that should be thoroughly addressed before it can be considered for publication:
- Why does the title of the manuscript end with a period?
- The research problem is not clearly stated — why do we need Msalais wine enriched in Amadori compounds?
- Line 193 – the dilution steps “10‑1” and “10‑7” should be written with superscripts.
- ITS1/ITS4 primers were used for molecular identification, but no GenBank accession numbers are provided for the obtained sequences.
- There is no information on the storage conditions of the isolated strains.
- The phylogenetic tree is of poor quality and difficult to interpret in its current version.
- Table 5 – “experimental” should be capitalized. Moreover, the note “Note: ‘+’ indicates gas production, and the more ‘+’ indicates the greater gas production.” is vague. The scale should be clearly defined (e.g., +, ++, +++ with specific criteria).
- Statistical analyses (e.g., ANOVA, Tukey’s test) are missing – although the authors describe changes in parameters, it is unclear whether the differences are statistically significant.
- The interpretation of the results lacks references to relevant literature – for instance, have similar Fru-Pro concentrations been observed in other fermented beverages?
- The discussion section often merely paraphrases the results without providing a deeper analysis of the underlying biological mechanisms.
- The conclusions lack practical reflection – is there any potential for scaling up or implementing the product in the food industry?
- Line 49: “Amadori compound is a key intermediate…” – this is a repetition of the same information already given in line 44.
- Line 268: “fermentation time (d)” – unclear: does this refer to full days or hours?
- Line 276: “placed in dark at room temperature for 2h” – the temperature range should be specified more precisely (e.g., 22–25 °C).
- Line 307: “the peak time of Fru-Pro and Fru-Asp was…” – this is grammatically incorrect; it should be “were” (plural).
- Line 323: “9.1595 min was Fru-Pro…” – this is colloquial and unscientific language; revise to: “The retention time of Fru-Pro was 9.16 min.”
Author Response
Comments 1: Why does the title of the manuscript end with a period?
Response 1: Thank you for pointing this out, I agree with this comment,so I have modified and marked it in line 2.
Comments 2: The research problem is not clearly stated — why do we need Msalais wine enriched in Amadori compounds?
Response 2: Thank you for pointing this out, I agree with this comment, so I have modified and marked it in line 50.
Comments 3: Line 193 – the dilution steps “101” and “107” should be written with superscripts.
Response 3: Thank you for pointing this out, I agree with this comment, so I have modified and marked it in line 281.
Comments 4: ITS1/ITS4 primers were used for molecular identification, but no GenBank accession numbers are provided for the obtained sequences.
Response 4: Thank you for pointing this out, I agree with this comment, so I have modified and marked it in line 300.
Comments 5: There is no information on the storage conditions of the isolated strains.
Response 5: Thank you for pointing this out, I agree with this comment, so I have modified and marked it in line 287.
Comments 6: The phylogenetic tree is of poor quality and difficult to interpret in its current version.
Response 6: Thank you for pointing this out, I agree with this comment, so I have modified and marked it in line 532.
Comments 7: Table 5 – “experimental” should be capitalized. Moreover, the note “Note: ‘+’ indicates gas production, and the more ‘+’ indicates the greater gas production.” is vague. The scale should be clearly defined (e.g., +, ++, +++ with specific criteria).
Response 7: Thank you for pointing this out, I agree with this comment, so I have modified and marked it in line 567.
Comments 8: Statistical analyses (e.g., ANOVA, Tukey’s test) are missing – although the authors describe changes in parameters, it is unclear whether the differences are statistically significant.
Response 8: Thank you for pointing this out, I agree with this comment, so I have modified and marked it in line 622 and line 694.
Comments 9: The interpretation of the results lacks references to relevant literature – for instance, have similar Fru-Pro concentrations been observed in other fermented beverages?
Response 9: Thank you for pointing this out, I agree with this comment, so I have modified and marked it in line 882.
Comments 10: The discussion section often merely paraphrases the results without providing a deeper analysis of the underlying biological mechanisms.
Response 10: Thank you for pointing this out, I agree with this comment, so I have modified and marked it in line 554, 570, 586, 601 and 612.
Comments 11: The conclusions lack practical reflection – is there any potential for scaling up or implementing the product in the food industry?
Response 11: Thank you for pointing this out, I agree with this comment, so I have modified and marked it in line 897.
Comments 12: Line 49: “Amadori compound is a key intermediate…” – this is a repetition of the same information already given in line 44.
Response 12: Thank you for pointing this out, I agree with this comment, so I have modified and marked it in line 66.
Comments 13: Line 268: “fermentation time (d)” – unclear: does this refer to full days or hours?
Response 13: Thank you for pointing this out, I agree with this comment, so I have modified and marked it in line 396.
Comments 14: Line 276: “placed in dark at room temperature for 2h” – the temperature range should be specified more precisely (e.g., 22–25 °C).
Response 14: Thank you for pointing this out, I agree with this comment, so I have modified and marked it in line 272, 290, 401, 412 and 414.
Comments 15:Line 307: “the peak time of Fru-Pro and Fru-Asp was…” – this is grammatically incorrect; it should be “were” (plural).
Response 15: Thank you for pointing this out, I agree with this comment, so I have modified and marked it in line 439.
Comments 16:Line 323: “9.1595 min was Fru-Pro…” – this is colloquial and unscientific language; revise to: “The retention time of Fru-Pro was 9.16 min.”
Response 16: Thank you for pointing this out, I agree with this comment, so I have modified and marked it in line 439 and 487.
Reviewer 2 Report
Comments and Suggestions for Authors
The article is interesting and addresses a relevant topic regarding the enrichment of Msalais wine with Amadori compounds through yeast fermentation. The idea of developing functional wines with enhanced antioxidant properties aligns well with current trends in food science and consumer health awareness. However, the study`s design, interpretation, and presentation require clarification and improvement before publication.
- The introduction provides useful background on Maillard chemistry, but it lacks focus and fails to clearly define the scope of the study. While the description of Amadori chemistry is informative, the introduction does not clearly state which compounds will be analyzed or justify their selection. Additionally, some sentences are repetitive or unclear (e.g., lines 44–57). The introduction would benefit from better organization, language polishing, and inclusion of a clear rationale for compound selection and analytical approach.
- The manuscript only analyzes two Amadori compounds, Fru-Pro and Fru-Asp, yet no justification is provided for this limited focus. Considering that the Maillard reaction can yield a wide variety of Amadori compounds depending on amino acid and sugar availability, the authors should explain why these two were chosen specifically.
- The manuscript should acknowledge the study`s limitations and suggest future directions. Since only two Amadori compounds were analyzed and no volatile or late-stage Maillard products were included, the authors should address these limitations. Future work could include more comprehensive profiling of Amadori compounds using untargeted approaches, such as HPLC-MS or NMR, as well as exploration of potential off-flavor or toxic by-products.
- The number of replicates and experimental repetitions is unclear and must be specified in order to assess the reproducibility and robustness of the results. Although three parallel experiments are mentioned, it is not clear whether these represent biological or technical replicates, nor how many independent fermentations were performed per condition. This information should be clarified.
- The reported increases in Fru-Pro and Fru-Asp content are modest and require appropriate statistical analysis for confirmation. The authors report increases in Fru-Pro and Fru-Asp content in optimized samples compared to commercial samples: Fru-Pro increased from 0.2190 to 0.2980 g/L, and Fru-Asp increased from 0.0176 to 0.0196 g/L. While the relative increase in Fru-Pro may be meaningful, the absolute differences, especially for Fru-Asp, are quite small. It is essential to include appropriate statistical analyses (e.g., t-tests, ANOVA) to confirm the significance of these differences.
- The antioxidant capacity data lack appropriate controls and contextual comparisons. Although values such as 116.37 μmol Trolox/sample (DPPH) and 142.51 μmol Trolox/sample (ABTS) are reported, comparisons with non-optimized Msalais or other relevant controls are missing, limiting interpretation.
- Several factual and typographical errors must be corrected for clarity and professionalism. For example, “Shimadzu Corporation (japan)” should be capitalized as “Japan.” The term “Mousalais” appears instead of “Msalais” (e.g., line 329). Inconsistent capitalization is evident in figure and table titles (e.g., “Table 4. microscopic morphology description table”), which should follow standard conventions. Scientific names of microorganisms, such as Saccharomyces cerevisiae or Wickerhamomyces anomalus, are not consistently italicized and sometimes incorrectly capitalized or abbreviated. All genus and species names should be italicized and follow standard taxonomic rules. Language should also be edited for clarity and scientific tone — for instance, phrases like “pleasant in fragrance” or “fermentation broth odor” are awkward and should be revised.
- Some citations are improperly formatted or incomplete and should be corrected.
To meet publication standards, the manuscript needs major revisions, including clarification of compound selection, statistical validation of results, improved methodological transparency, and correction of typographical and language errors.
Author Response
Comments 1:The introduction provides useful background on Maillard chemistry, but it lacks focus and fails to clearly define the scope of the study. While the description of Amadori chemistry is informative, the introduction does not clearly state which compounds will be analyzed or justify their selection. Additionally, some sentences are repetitive or unclear (e.g., lines 44–57). The introduction would benefit from better organization, language polishing, and inclusion of a clear rationale for compound selection and analytical approach.
Response 1: Thank you for pointing this out, I agree with this comment, so I have modified and marked it in line 66.
Comments 2: The manuscript only analyzes two Amadori compounds, Fru-Pro and Fru-Asp, yet no justification is provided for this limited focus. Considering that the Maillard reaction can yield a wide variety of Amadori compounds depending on amino acid and sugar availability, the authors should explain why these two were chosen specifically.
Response 2: Thank you for pointing this out, I agree with this comment, so I have modified and marked it in line 57.
Comments 3: The manuscript should acknowledge the study`s limitations and suggest future directions. Since only two Amadori compounds were analyzed and no volatile or late-stage Maillard products were included, the authors should address these limitations. Future work could include more comprehensive profiling of Amadori compounds using untargeted approaches, such as HPLC-MS or NMR, as well as exploration of potential off-flavor or toxic by-products..
Response 3: Thank you for pointing this out, I agree with this comment, so I have modified and marked it in line 913.
Comments 4: The number of replicates and experimental repetitions is unclear and must be specified in order to assess the reproducibility and robustness of the results. Although three parallel experiments are mentioned, it is not clear whether these represent biological or technical replicates, nor how many independent fermentations were performed per condition. This information should be clarified.
Response 4: Thank you for pointing this out, I agree with this comment, so I have modified and marked it in line 345-389.
Comments 5: The reported increases in Fru-Pro and Fru-Asp content are modest and require appropriate statistical analysis for confirmation. The authors report increases in Fru-Pro and Fru-Asp content in optimized samples compared to commercial samples: Fru-Pro increased from 0.2190 to 0.2980 g/L, and Fru-Asp increased from 0.0176 to 0.0196 g/L. While the relative increase in Fru-Pro may be meaningful, the absolute differences, especially for Fru-Asp, are quite small. It is essential to include appropriate statistical analyses (e.g., t-tests, ANOVA) to confirm the significance of these differences.Response 5: Thank you for pointing this out, I agree with this comment, so I have modified and marked it in line 625 and 697.
Comments 6: The antioxidant capacity data lack appropriate controls and contextual comparisons. Although values such as 116.37 μmol Trolox/sample (DPPH) and 142.51 μmol Trolox/sample (ABTS) are reported, comparisons with non-optimized Msalais or other relevant controls are missing, limiting interpretation.
Response 6: Thank you for pointing this out, I agree with this comment, so I have modified and marked it in line 30, 833 and 871.
Comments 7: Several factual and typographical errors must be corrected for clarity and professionalism. For example, “Shimadzu Corporation (japan)” should be capitalized as “Japan.” The term “Mousalais” appears instead of “Msalais” (e.g., line 329). Inconsistent capitalization is evident in figure and table titles (e.g., “Table 4. microscopic morphology description table”), which should follow standard conventions. Scientific names of microorganisms, such as Saccharomyces cerevisiae or Wickerhamomyces anomalus, are not consistently italicized and sometimes incorrectly capitalized or abbreviated. All genus and species names should be italicized and follow standard taxonomic rules. Language should also be edited for clarity and scientific tone — for instance, phrases like “pleasant in fragrance” or “fermentation broth odor” are awkward and should be revised.
Response 7: Thank you for pointing this out, I agree with this comment, so I have modified and marked it in line162, 506, 529 and 569.
Comments 8: Some citations are improperly formatted or incomplete and should be corrected.
Response 8: Thank you for pointing this out, I agree with this comment, so I have modified and marked it in line 942 and 955.
Reviewer 3 Report
Comments and Suggestions for Authors
Introduction
1. The introduction discusses several analytical techniques for detecting Amadori compounds. However, the method actually used in the study (HPLC with evaporative light scattering detector, ELSD) is not introduced or discussed in this context. So, introduce ELSD explicitly and justify its suitability for detecting Amadori compounds.
2. Several acronyms are used without being properly defined on first mention, and in some cases the full term is never revisited again.
3. Expand the reference list and ensure that every major scientific claim (especially health-related or methodological) is appropriately supported by peer-reviewed literature.
Results and discussion
Section 3.1 claims to validate an HPLC-ELSD method for detecting Fru-Asp and Fru-Pro, but the validation is incomplete and does not follow any established method validation guidelines (e.g., ICH Q2(R1), AOAC, FDA).
- The method lacks key validation parameters such as LOD, LOQ, accuracy, precision, recovery, robustness, and specificity.
- The calibration curve range (100–500 µg/mL) is not appropriate for Fru-Asp, which was found at ~17.6 µg/mL — below the calibration range. This invalidates the quantification of Fru-Asp unless a lower calibration point or sensitivity analysis is added.
- Sentences such as “The separation of the two substances was better” are vague and non-scientific. "Better" than what? This is not comparative.
- The validity of the entire study relies on accurate, reliable detection and quantification of Amadori compounds (Fru-Asp and Fru-Pro) in Msalais samples. However, because the analytical method was not properly validated, the credibility of all subsequent experimental results — including comparisons between samples, fermentation optimization, and antioxidant analysis — is severely compromised.
- Table 8 lacks the units for Fru-Pro and Fru-Asp concentrations. This omission prevents proper interpretation of the data. Moreover, if the values are in g/L as implied earlier, the Fru-Asp values are mostly below the validated calibration range (0.1–0.5 g/mL). This indicates the use of the analytical method outside its validated dynamic range, violating basic principles of method validation and rendering the reported concentrations unreliable.
Author Response
Comments 1: The introduction discusses several analytical techniques for detecting Amadori compounds. However, the method actually used in the study (HPLC with evaporative light scattering detector, ELSD) is not introduced or discussed in this context. So, introduce ELSD explicitly and justify its suitability for detecting Amadori compounds.
Response 1: Thank you for pointing this out, I agree with this comment, so I have modified and marked it in line 118.
Comments 2: Several acronyms are used without being properly defined on first mention, and in some cases the full term is never revisited again.
Response 2: Thank you for pointing this out, I agree with this comment, so I have modified and marked it in line 112 and 117.
Comments 3: Expand the reference list and ensure that every major scientific claim (especially health-related or methodological) is appropriately supported by peer-reviewed literature.
Response 3: Thank you for pointing this out, I agree with this comment, so I have modified and marked it in line 53, 112, 118, 279 and 888.
Comments 4: Section 3.1 claims to validate an HPLC-ELSD method for detecting Fru-Asp and Fru-Pro, but the validation is incomplete and does not follow any established method validation guidelines (e.g., ICH Q2(R1), AOAC, FDA).
Response 4: Thank you for pointing this out, I agree with this comment, so I have modified and marked it in line 205
Comments 5: The method lacks key validation parameters such as LOD, LOQ, accuracy, precision, recovery, robustness, and specificity.
Response 5: Thank you for pointing this out, I agree with this comment, so I have modified and marked it in line 206-263 and 436-504
Comments 6: The calibration curve range (100–500 µg/mL) is not appropriate for Fru-Asp, which was found at ~17.6 µg/mL — below the calibration range. This invalidates the quantification of Fru-Asp unless a lower calibration point or sensitivity analysis is added.
Response 6: Thank you for pointing this out, I agree with this comment, so I have modified and marked it in line 447-462
Comments 7: Sentences such as “The separation of the two substances was better” are vague and non-scientific. "Better" than what? This is not comparative.
Response 7: Thank you for pointing this out, I agree with this comment, so I have modified and marked it in line 493.
Comments 8: The validity of the entire study relies on accurate, reliable detection and quantification of Amadori compounds (Fru-Asp and Fru-Pro) in Msalais samples. However, because the analytical method was not properly validated, the credibility of all subsequent experimental results — including comparisons between samples, fermentation optimization, and antioxidant analysis — is severely compromised.
Response 8: Thank you for pointing this out, I agree with this comment, so I have modified and marked it in line 436-504.
Comments 9: Table 8 lacks the units for Fru-Pro and Fru-Asp concentrations. This omission prevents proper interpretation of the data. Moreover, if the values are in g/L as implied earlier, the Fru-Asp values are mostly below the validated calibration range (0.1–0.5 g/mL). This indicates the use of the analytical method outside its validated dynamic range, violating basic principles of method validation and rendering the reported concentrations unreliable.
Response 9: Thank you for pointing this out, I agree with this comment, so I have modified and marked it in line 460 and 713.